# Visual motion perception as online hierarchical inference

Johannes Bill [1,2] ✉, Samuel J. Gershman [2,3,4,5] & Jan Drugowitsch [1,3,5]

Identifying the structure of motion relations in the environment is critical for navigation, tracking, prediction, and pursuit. Yet, little is known about the mental and neural computations that allow the visual system to infer this structure online from a volatile stream of visual information. We propose online hierarchical Bayesian inference as a principled solution for how the brain might solve this complex perceptual task. We derive an online Expectation-Maximization algorithm that explains human percepts qualitatively and quantitatively for a diverse set of stimuli, covering classical psychophysics experiments, ambiguous motion scenes, and illusory motion displays. We thereby identify normative explanations for the origin of human motion structure perception and make testable predictions for future psychophysics experiments. The proposed online hierarchical inference model furthermore affords a neural network implementation which shares properties with motion-sensitive cortical areas and motivates targeted experiments to reveal the neural representations of latent structure.

Efficient behavior requires identification of structure in a continuous stream of volatile and often ambiguous visual information. To identify this structure, the brain exploits statistical relations in velocities of observable features, such as the coherent motion of features composing an object (Fig. 1a). Motion structure thus carries essential information about the spatial and temporal evolution of the environment, and aids behaviors such as navigation, tracking, prediction, and pursuit[1–8]. It remains, however, unclear how the visual system identifies a scene's underlying motion structure and exploits it to turn noisy, unstructured, sensory impressions into meaningful motion percepts.

In recent years, Bayesian inference has provided a successful normative perspective on many aspects of visual motion perception[9–17]. Human perception of motion stimuli spatially constrained by an aperture is well-explained by Bayesian statistical inference[9–11,14], and neural circuits that integrate local retinal input into neural representations of motion have been identified[18–23]. For the perception of structured motion spanning multiple objects and larger areas of the visual field, however, a comprehensive understanding is only beginning to emerge[15,24–27]. While common fate, that is, the use of motion coherence for grouping visual features into percepts of rigid

objects, received some experimental support[24,28], the perception of natural scenes requires more flexible structure representations (e.g., nested motion relations and non-rigid deformations) than common fate alone. Recent theoretical work[15] has introduced a representation of tree structures for the mental organization of observed velocities into nested hierarchies. Theory-driven experiments subsequently demonstrated that the human visual system indeed makes use of hierarchical structure when solving visual tasks[16], and that salient aspects of human motion structure perception can be explained by normative models of Bayesian inference over tree structures[17]. Because these studies were restricted to modeling motion integration only with regard to the perceptual outcome—they analyzed presented visual scenes offline using ideal Bayesian observer models—it remained unclear how the visual system solves the chicken-and-egg problem of parsing (in real time) instantaneous motion in a scene while simultaneously inferring the scene's underlying structure.

We address this question by formulating visual motion perception as online hierarchical inference in a generative model of structured motion. The resulting continuous-time model is able to explain human perception of motion stimuli covering classical psychophysics

[1]Department of Neurobiology, Harvard Medical School, Boston, MA, USA. [2]Department of Psychology, Harvard University, Cambridge, MA, USA. [3]Center for Brain Science, Harvard University, Cambridge, MA, USA. [4]Center for Brains, Minds, and Machines, MIT, Cambridge, MA, USA. [5]These authors jointly supervised this work: Samuel J. Gershman, Jan Drugowitsch. ✉e-mail: johannes_bill@hms.harvard.edu

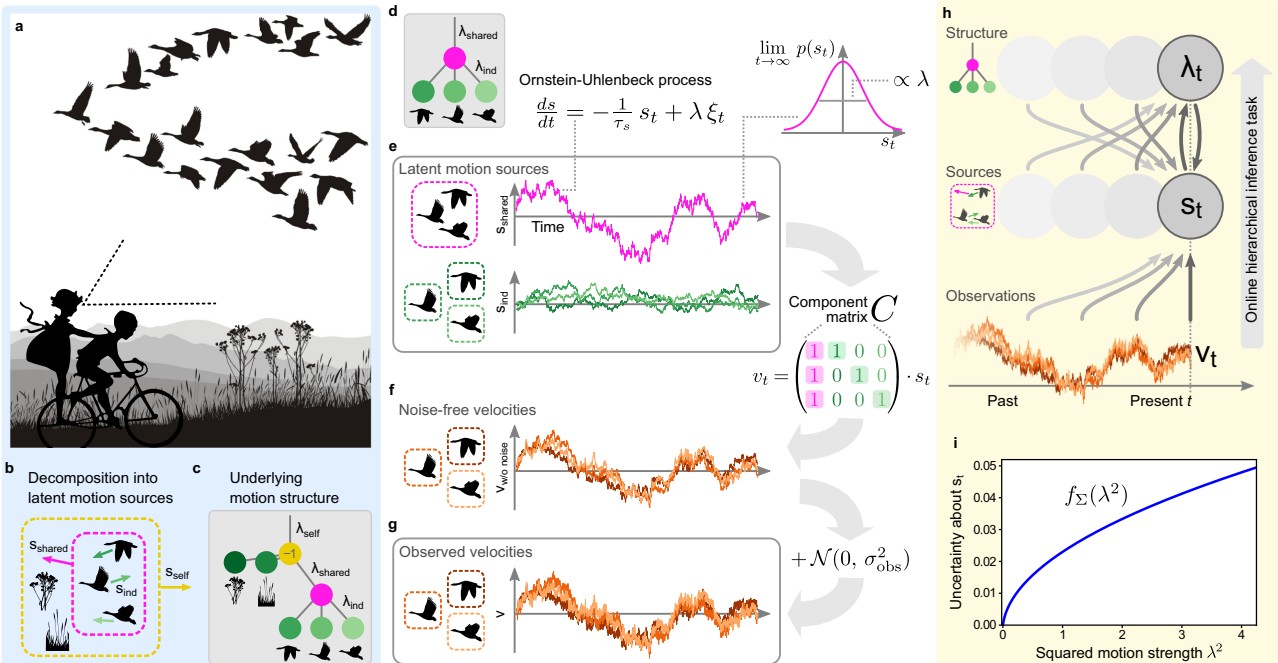

**Fig. 1 | Visual motion perception as an online hierarchical inference task.**
**a** Scene with nested motion relations. Observed velocities reaching the observer's retina are perceived as a combination of self-motion, flock motion and every bird's individual motion relative to the flock. **b** Formal decomposition of the scene's motion into latent motion sources. **c** Tree-structured graph representation of the underlying motion structure with nodes corresponding to latent motion sources. Self-motion contributes in the opposite direction to retinal velocity (−1). Vertical distances between nodes, termed motion strengths, $\lambda$, describe the long-term average speed of the source. Vanishing motion strength indicates that the corresponding motion source is not present the scene. **d**–**g** Generative model of structured motion. **d** Graph for a simpler motion scene with three flocking birds and a stationary observer.

**e** Latent motion sources follow independent Ornstein–Uhlenbeck processes. **f** The component matrix, $C$, composes noise-free velocities from the motion sources, such that each velocity is the sum of all its ancestral sources. **g** Observed velocities are noisy versions of the noise-free velocities. **h** Inverting the generative model according to Bayes' rule poses an online hierarchical inference task characterized by interdependent updates of motion sources and structure. **i** Using an adiabatic approximation, the motion sources' posterior variances reduce to a function of the motion strengths. Panels **a**–**h** are derived from artwork by Vladimír Čerešňák ("Migrating geese in the spring and autumn" licensed from Depositphotos Inc.) and Gordon Dylan Johnson ("Vintage Brother And Sister Bicycle Silhouette" from Openclipart.org, public domain).

experiments, ambiguous motion scenes, and illusory motion displays. The model, which relies on online Expectation-Maximization[29–31], separates inference of instantaneous motion from identifying a scene's underlying structure by exploiting the fact that these evolve on different time-scales. The resulting set of interconnected differential equations decomposes a scene's velocities with the goal of minimizing prediction errors for subsequent observations. Beyond capturing human percepts in many psychophysics experiments qualitatively, the model explains human motion structure classification quantitatively with higher fidelity than a previous ideal observer-based model[17]. Furthermore, the model provides a normative explanation for the putative origin of human illusory motion perception, and yields testable predictions for future psychophysics experiments.

Finally, we address how motion structure discovery could be supported by neural circuits in the brain. Studying the neural representations underlying motion structure perception is challenging, as the perceived structure often has no direct physical counterpart in the environment (e.g., the concept of a flock velocity in Fig. 1a). We derive a recurrent neural network model that not only implements the proposed online hierarchical inference model, but shares many properties with motion-sensitive middle temporal area (MT)[21] and dorsal medial superior temporal area (MSTd)[19,32]. The network model in turn allows us to propose a class of stimuli for neuroscientific experiments that make concrete predictions for neural recordings.

## Results
In what follows, we first present the online model for simultaneous hierarchical inference of instantaneous motion and of the scene's underlying structure. Next, we demonstrate the model's ability to

explain human motion perception across a set of psychophysics experiments and discuss testable predictions for future studies. Finally, we propose a biologically realistic neural implementation of online hierarchical inference and identify targeted experiments to reveal neural representations of latent structure.

## Online hierarchical inference in a generative model of structured motion
A structural understanding of the scene in Fig. 1a requires the observer to decompose observed velocities of objects or their features into what we call latent motion sources, $s$, that, together, compose the scene (Fig. 1b). These latent sources might or might not have a direct counterpart in the physical world. In Fig. 1b, for instance, each bird's velocity on the observer's retina can be decomposed into the observer's self-motion, $s_{self}$, the flock's motion, $s_{shared}$, plus a smaller, animal-specific component, $s_{ind}$. Here, flock motion is an abstract mental concept that is introduced to organize perception, but doesn't have an immediate physical correlate. A correct decomposition leads to motion sources that aid interpretation of the visual scene, and thus supports behaviors such as navigation, tracking, prediction and pursuit. Such decomposition requires knowledge of the scene's structure, like the presence of a flock and which birds it encompasses (Fig. 1c). Wrong structural assumptions might lead to faulty inference of motion sources, like wrongly attributing the flock's motion in the sky to self-motion. Thus, the challenge for an observer is to simultaneously infer motion sources and structure online from a stream of noisy and ambiguous visual information.

We formalized the intuition of structured motion in the generative model shown in Fig. 1d–g. The stochastic model, first

introduced in ref. 16 and formally defined in Supplementary Note 1, accommodates fundamental principles of physics (isotropy and inertia) and psychophysics (continuity of trajectories[33] and slow-velocity priors[9]), without making assumptions on specific object trajectories. For example, the motion of three flocking birds viewed by a stationary observer (motion tree in Fig. 1d) can be decomposed into four independent motion sources—one shared (magenta) and three individual (green, one per bird)—that evolve according to Ornstein–Uhlenbeck processes[34], generating smooth motion with changes typically occurring at time scale $\tau_s$ (Fig. 1e). The resulting speed (absolute velocity) distribution of each motion source is governed by an associated motion strength, $\lambda$, such that the expected speed is proportional to $\lambda$. The observable velocities, $v_t$, are in turn noise-perturbed (noise magnitude $\sigma_{\mathrm{obs}}$; Fig. 1g) sums of the individual motion sources (collected in vector $s_t$), with the contribution of each individual motion source specified by a different column of the component matrix $C$ (see Fig. 1f). This formalizes the intuition that observable velocities are the sum of their ancestral motion sources in the tree.

In this model, the structure of a scene is fully characterized by the vector of motion strengths, $\lambda = (\lambda_1, \ldots, \lambda_m, \ldots, \lambda_M)$, which describe the presence ($\lambda_m > 0$) or absence ($\lambda_m = 0$) of motion components, as well as their typical speed. In other words, given a reservoir of components, $C$, which might have been learned to occur in visual scenes in general, knowing $\lambda$ is equivalent to knowing the motion structure of the scene. Inferring this structure in turn becomes equivalent to inferring the corresponding motion strengths.

An agent faces two challenges when performing inference in this generative model (Fig. 1h). First, inference needs to be performed on the fly (i.e., online) while sensory information arrives as an ongoing stream of noisy velocity observations. Second, how observed motion is separated into latent motion sources, $s$, and motion structure, $\lambda$, is inherently ambiguous, such that inference needs to resolve the hierarchical inter-dependence between these two factors. We address both challenges by recognizing that motion structure, $\lambda$, typically changes more slowly than the often volatile values of motion sources, $s$, facilitating the use of an online Expectation-Maximization (EM) algorithm to infer both. This separation of time scales yields a system of mutually dependent equations for updating $\lambda$ and $s$ and furthermore affords a memory-efficient, continuous-time online formulation that is amenable to a neural implementation (see Methods for an outline of the derivation, and Supplementary Note 2 for the full derivation). While the algorithm is approximate, it nonetheless performs adequate online hierarchical inference and closely resembles more accurate solutions, even for deeply nested motion structures (see Supplementary Fig. 1).

Our online model computes, at any time, a posterior belief over the latent motion sources, $s_t$, which is Gaussian with mean vector $\mu_t$ and covariance matrix $\Sigma_t$, as well as an estimate, $\lambda_t$, of the underlying structure. The dynamics of $\mu_t$, $\Sigma_t$, and $\lambda_t^2$ (the inference is more elegantly formulated on the squared values) read:

$$\partial_t \lambda_t^2 = -\frac{1}{\tau_\lambda} \lambda_t^2 + \alpha \odot \left( \mu_t^2 + f_\Sigma(\lambda_t^2) \right) + \beta, \tag{1}$$

$$\partial_t \mu_t = -\frac{1}{\tau_s} \mu_t + f_\Sigma(\lambda_t^2) \odot C^\top \epsilon_t \text{ with } \epsilon_t = \frac{v_t}{\sigma_{\mathrm{obs}}^2} - \frac{C \mu_t}{\sigma_{\mathrm{obs}}^2}, \tag{2}$$

$$\text{and } \Sigma_t = \mathrm{diag}\left[ f_\Sigma(\lambda_t^2) \right]. \tag{3}$$

The coupled Eqs. (1)–(3) support the following intuition. Equation (1) calculates a running average of the motion strengths $\lambda_t^2$ by use of a low-pass filter with time scale $\tau_\lambda$. Here, $\odot$ denotes element-wise multiplication and the function $f_\Sigma(\lambda_t^2)$ (Fig. 1i) estimates the variance of the $s$-posterior distribution according to an adiabatic approximation (cf. Eq. (3), see Methods). The constants $\alpha$ and $\beta$ contribute a sparsity-

promoting prior, $p(\lambda^2)$, for typical values of the motion strengths (see Methods for their full expressions). By Eq. (2), the motion source means $\mu_t$ are estimated by a slightly different low-pass filter that relies on a prediction error, $\epsilon_t$, between the model's expected velocities, $C\mu_t$, and those actually encountered in the input, $v_t$ (both normalized by observation noise variance to facilitate the later network implementation). This prediction error on observable velocities is transformed back to the space of latent motion sources via the transposed component matrix $C^\top$ and then, importantly, gated by element-wise multiplication ($\odot$) with the variance estimates $f_\Sigma(\lambda_t^2)$. This gating implements a credit assignment as to which motion source was the likely cause of observed mismatches in $\epsilon_t$, and thus uses the scene's currently inferred motion structure to modulate the observed velocities' decomposition into motion sources. For flocking birds, for example, a simultaneous alignment in multiple birds' velocities would only be attributed to the shared flock velocity if such a flock had been detected in the past ($\lambda_{\mathrm{shared}}$ large, and $\lambda_{\mathrm{ind}}$ small). Otherwise it would be assigned to the birds' individual motions, $s_{\mathrm{ind}}$.

Together, Eqs. (1) and (2) implement a coupled process of structure discovery and motion decomposition, which distinguishes them through different time-scales. Notably, the proposed model is not a heuristic, but is derived directly from a normative theory of online hierarchical inference. Next, we explored if the model can explain prominent phenomena of human visual motion perception.

## Online inference replicates human perception of classical motion displays

To explore if the proposed online model can qualitatively replicate human perception of established motion displays, we simulated two classical experiments from Gunnar Johansson[25] and Karl Duncker[35]. These experiments belong to a class of visual stimuli which we refer to as object-indexed experiments (Fig. 2a) because the observed velocities, $v_t$, belong to objects irrespective of their spatial locations. (A second class, which we refer to as location-indexed experiments, will be discussed below.)

In Johansson's experiment, three dots oscillate about the screen with two of the dots moving horizontally and the third dot moving diagonally between them (see Fig. 2b and Supplementary Movie 1). Humans perceive this stimulus as a shared horizontal oscillation of all three dots, plus a nested vertical oscillation of the central dot. Similar to previous offline algorithms[15], our online model identifies the presence of two motion components (Fig. 2c): a strong shared motion strength, $\lambda_{\mathrm{shared}}$ (magenta) and weaker individual motion, $\lambda_{\mathrm{ind}}$, for the central dot (green). The individual strengths of the outer two dots (light and dark green), in contrast, decay to zero. Most motion sources within the structure are inferred to be small (dotted lines in Fig. 2d). Only two sources feature pronounced oscillations: the x-direction of the shared motion source, $\mu_{\mathrm{shared},x}$, (magenta, solid line) and the y-direction of the central dot's individual source, $\mu_{\mathrm{ind},y}$, (green, solid line), mirroring human perception. As observed velocities are noisy, they introduce noise in the inferred values of $\mu_t$, which fluctuate around the smooth sine-functions of the original, noise-free stimulus. As expected from well-calibrated Bayesian inference, the magnitude of these fluctuations is correctly mirrored in the model's uncertainty, as illustrated by the posteriors' standard deviation $\sqrt{f_\Sigma(\lambda_t^2)}$ (shaded areas in Fig. 2d).

In the second experiment, known as the Duncker wheel, two dots follow the motion of a rolling wheel, one marking the hub, the other marking a point on the rim (Fig. 2e). The two dots describe an intricate trajectory pattern (see Fig. 2f and Supplementary Movie 2), that, despite its impoverished nature, creates the impression of a rolling object for human observers, a percept that has been replicated by offline algorithms[15]. Likewise, our online model identifies a shared (magenta in Fig. 2g) plus one individual (dark green) component, and decomposes the observed velocities into shared rightward motion

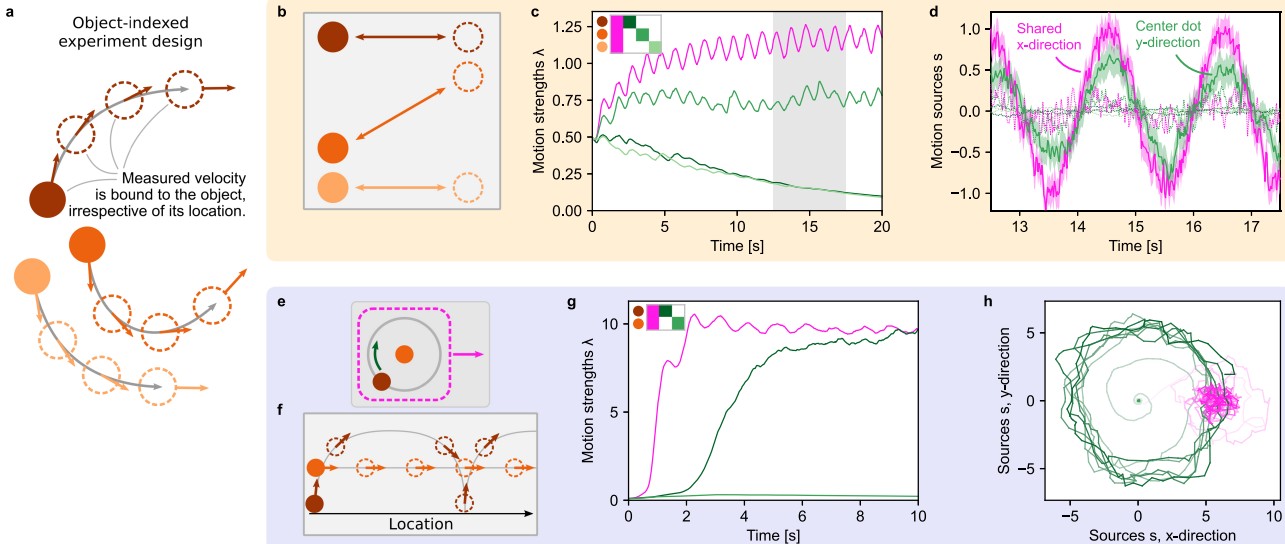

**Fig. 2 | Online hierarchical inference replicates human perception of classical motion displays. a** In object-indexed experiment designs, every observable velocity is bound to an object irrespective of its location. Many psychophysics studies fall into this class of experiment design. **b** Johansson's 3-dot motion display. Humans perceive the stimulus as shared horizontal motion with the central dot oscillating vertically between the outer dots. **c** The online model's estimate of the motion strengths, $\lambda_t$ (a single motion strength is shared across both spatial dimensions). The component matrix, $\boldsymbol{C}$, is shown in the top-left as a legend for the line colors. Circles next to the matrix show the assignment of the rows in $\boldsymbol{C}$ to the dots in panel **b**. **d** The model's posterior distribution over the motion sources, $\boldsymbol{s}_t$, during the gray-shaded period in panel **c**. Shown are the mean values, $\boldsymbol{\mu}_t$, as lines along with the model's estimated standard deviation (shaded, only for two components for visual clarity). **e** The Duncker wheel resembles a rolling wheel of which only the hub and one dot on the rim are visible. **f** Despite its minimalist trajectory pattern, humans perceive a rolling wheel. **g** Inferred motion strengths, $\lambda_t$. The model identifies shared motion plus an individual component for the revolving dot. **h** Inferred motion sources, $\boldsymbol{\mu}_t$, for the duration in panel **g**. Color gradients along the lines indicate time (from low to high contrast). For visual clarity, $\boldsymbol{\mu}_t$ has been smoothed with a 50 ms box filter for plotting. Source data are provided as a Source Data file.

plus rotational motion for the dot on the rim (see Fig. 2h). Notably, the shared motion component is discovered before the revolving dot's individual motion, leading to a transient oscillation in the inferred shared motion source, $\mu_{\text{shared}}$ (see light magenta trace in Fig. 2h) — an onset effect that could be tested experimentally.

In summary, the online hierarchical inference model successfully identified the structure underlying the motion displays, provided Bayesian certainty estimates for the inferred motion, and replicated human perception in these classical psychophysics experiments.

## Online inference outperforms ideal observers in explaining human structure perception

Having qualitatively replicated motion structure inference in common motion displays, we next asked if our online model could quantitatively explain human motion structure perception. To address this question, we reevaluated behavioral data from Yang et al.[17], where participants had to categorize the latent structure of short motion displays (see Fig. 3a). Motion scenes followed one of four structures (Fig. 3b) and were generated stochastically from the same generative model underlying our hierarchical inference model. Owing to their stochastic generation, scenes often were ambiguous with regard to their latent structure, prompting distinct error patterns in human responses (see confusion matrix in Fig. 3c). For instance, independently moving dots were more frequently misclassified as clustered motion (I-C element) than vice versa (C-I element), global motion was highly recognizable, and nested hierarchical motion was more frequently misperceived as clustered than as global.

To test if human responses arise from normative, Bayesian motion structure inference, Yang et al. modeled these responses in two steps (blue branch in Fig. 3d): first, an offline Bayesian ideal observer, which was provided with the trajectories of all objects within a trial, calculated the likelihood for each of the four structures. Then, these four probabilities were fed into a choice model with a small set of participant-specific fitting parameters (see Methods). This model

captured many aspects of human responses, including task performance, typical error patterns, single-trial responses, and participant-specific differences. Yet, the model arrived at these probabilities by comparing the likelihoods of the full sequences for all four candidate structures, and so had no notion of how a percept of structure could emerge over the course of the trial.

Thus, we next asked if our online model, which gradually infers the structure during the stimulus presentation, was better able to account for the observed response pattern. As our model by design inferred real-valued motion strengths $\lambda$ rather than only discriminating between the four structures used in the experiment, we added an additional stage that turned the inferred motion strengths into a likelihood for each of the four structures at trial end (red branch in Fig. 3d, see Methods). To do so, we computed five hand-designed features from the seven-dimensional vector $\lambda_t$ (besides one global and three individual strengths, there are three possible two-dot clusters), and trained a multinomial logistic regression classifier on the features to obtain likelihood values for each of the structures. The classifier was trained on the true structures of the trials, and thus contained no information about human responses. Finally, we fitted the same choice model as Yang et al. to the participants' responses.

The confusion matrix predicted by our model shows an excellent agreement with human choices, both when averaged across participants (Fig. 3e), and on a per-participant basis (see Supplementary Figs. 3 and 4). Indeed, our model beats the original computational model in terms of response log-likelihoods for all of the 12 participants (see Fig. 3f; $p < 0.001$, two-sided paired Wilcoxon signed-rank test). Furthermore, the online model overcomes the systematic underestimation of global motion (G-G matrix element) that previous, ideal observer-based approaches suffered from[16,17]. Importantly, in our model, any information connecting the stimulus to the eventual choice is conveyed through the motion strengths, $\lambda_t$, as a bottleneck. The fact that the online hierarchical inference-based approach describes human responses better than the ideal observer-based model of Yang

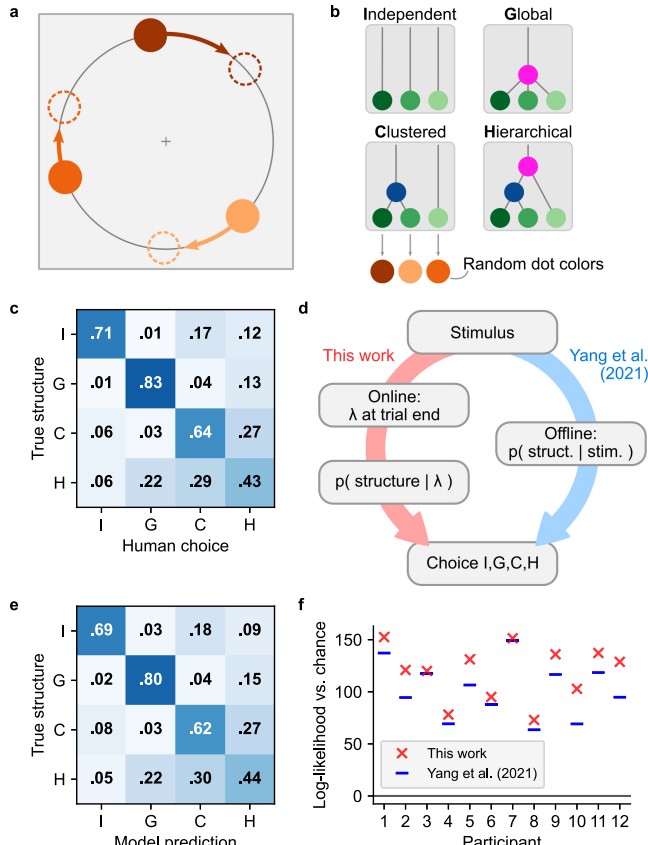

**Fig. 3 | The model quantitatively explains human perception of nested and ambiguous motion scenes. a** Stochastic motion stimulus from Yang et al.[17] consisting of three dots rotating on a circle. **b** Each trial followed one of four motion structures. If clustered motion was present (C or H structure), any pair of dots could form the cluster. **c** Confusion matrix of human responses, averaged over all 12 participants. **d** Models for predicting human responses. Yang et al. employed a Bayesian ideal observer as the basis for fitting a participant-specific choice model. Our model, in contrast, calculates the likelihood for each structure from the motion strengths, $\lambda_t$, at trial end and then fits the same choice model as Yang et al. for translating probabilities into human responses. **e** Confusion matrix of our model. **f** Log-likelihood of human responses relative to chance level, for both models. The analyses in panels **e** and **f** are leave-one-out cross-validated to prevent overfitting. Source data are provided as a Source Data file.

et al. indicates that our model may share mechanistic features with the human perceptual apparatus.

**Explaining motion illusions that rely on spatial receptive fields**
In contrast to the object-indexed experiments discussed above, another class of psychophysics experiments employs velocity stimuli that remain at stationary locations (see Fig. 4a), typically in the form of apertures of moving dots or drifting Gabors. This class, which we refer to as location-indexed experiments, is furthermore popular in neuroscience as it keeps the stimulus' local visual flow within an individual neuron's spatial receptive field throughout the trial[21]. We investigated our model's ability to explain illusory motion perception in two different types of location-indexed experiments: motion direction repulsion in random-dot kinematograms (RDKs)[36–41], see Fig. 4, and noise-dependent motion integration of spatially distributed stimuli[42,43], see Fig. 5.

We modeled perception in these experiments by including a self-motion component and added a vestibular input signal to the observables (see Fig. 4b, and cf. Fig. 1a–c). The vestibular input, which we fixed to have zero mean plus observation noise, complemented the visual input, which is ambiguous with regard to self-motion and

globally shared object motion and can induce illusory self-motion ("vection")[44–46]. In turn, we model the subjectively perceived velocity of objects, relative to the stationary environment, as the sum of all inferred motion sources excluding self-motion (see Fig. 4c and Methods).

In the RDK experiment, a participant fixates the center of an aperture in which two groups of randomly positioned dots move linearly with opening angle $\gamma$ (see Fig. 4d) and subsequently reports the perceived opening angle. Motion direction repulsion occurs if the perceived angle is systematically biased relative to the true opening angle.

As previously reported, the repulsion bias can change from an under-estimation of the opening angle for small angles to an over-estimation for large angles (data from ref. 36 reprinted as black dots in Fig. 4e). We replicated this effect by simulating two constant dot velocities with opening angles that varied across trials. Our model decomposed the stimulus into self-motion, shared motion and individual (group) motion. Across opening angles, it featured a triphasic psychometric function with angles smaller than ~40° being under-estimated, angles between ~40° and ~110° being over-estimated, and even larger angels being unbiased (purple curve in Fig. 4e). The match with human biases arose without systematic tuning of simulation parameters (the simulations presented in this manuscript were mostly performed with a set of default parameters, see Methods). Inspecting the model's inferred motion components revealed that, for small $\gamma$, the negative bias arose from integrating all dots into a single, coherent motion component while disregarding individual dot motions (left inset in Fig. 4e). Intermediate $\gamma$, in contrast, caused the shared component to be correctly broken up into two individual components—plus a small illusory self-motion component (right inset in Fig. 4e). This self-motion, which is ignored in the perceived velocities, widened the perceived opening angle between the two groups of dots. For even larger $\gamma$, the illusory self-motion vanished yielding unbiased percepts.

For fixed opening angles, motion direction repulsion is furthermore modulated by relative contrast and speed difference between the two motion components. Specifically, for an opening angle of $\gamma = 45°$, Chen et al.[37] have shown that increasing the contrast of one dot group inflates the perceived opening angle—here measured relative to horizontal to separate cause and effect—of the other, constant-contrast group (Fig. 4f, left). We replicated this effect in simulations that operationalized visual contrast as an (inverse) multiplicative factor on the observation noise variance, $\sigma_{obs}^2$. For an opening angle of $\gamma = 45°$, our model featured a positive and monotonically increasing repulsion bias as the second group's contrast increases (purple line in Fig. 4f, right), similar to what has been previously reported. For smaller opening angles, in contrast, our model predicts an inversion of the repulsion bias, which first decreases at low contrast and then increases again for higher contrast (blue line in Fig. 4f, right)—a prediction that remains to be tested. Increasing the speed of one motion component for large opening angles also introduces a positive bias in the perceived opening angle of the other component in human participants[36,38]. We replicated this effect by increasing the second group's speed, which, for a $\gamma = 90°$ opening angle, yielded a relatively stable bias of ~5° across different motion speeds (dashed line in Fig. 4g), in line with the aforementioned experimental data from Braddick et al.[36] and, for a $\gamma = 60°$ opening angle (purple line in Fig. 4g), qualitatively replicated the initial rise and then gradual decline in the bias, as reported for this opening angle by Benton and Curran[38]. Furthermore, our model predicts that the speed-dependent bias changes to a biphasic curve for smaller opening angles (blue line), providing another testable prediction.

Extending the basic MDR experiment from Fig. 4d, Takemura et al.[39] investigated how motion in a surrounding annulus affects the perceived directions of inner RDKs, see sketch in the top left of Fig. 4h. Two inner RDKs move to the left and right, respectively, while two

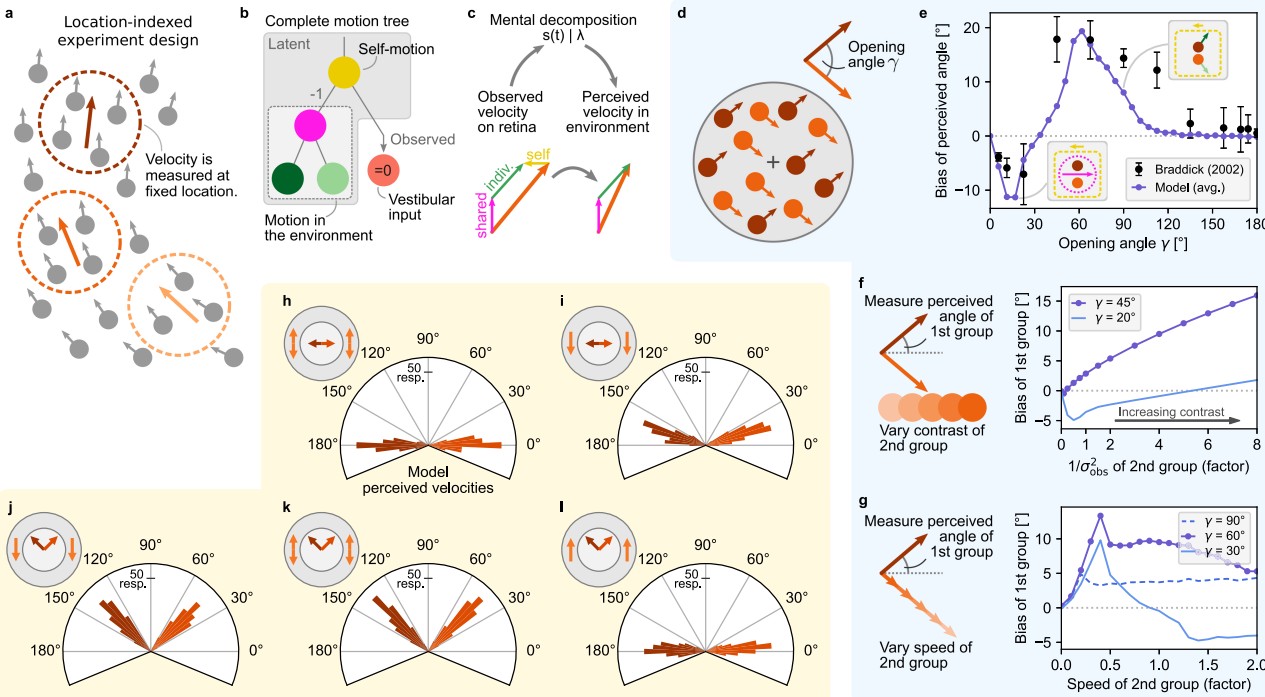

**Fig. 4 | Hierarchical inference explains motion illusions in location-indexed experiments. a** In location-indexed experiments, motion flow is presented at stationary spatial locations. **b** Considered latent motion components. Self-motion, which affects all retinal velocities in the opposite direction (−1) integrates both visual input and a vestibular signal (here: zero + noise). **c** Perceived object velocities, relative to the environment, are the sum of all inferred motion components excluding self-motion. **d** In motion direction repulsion experiments, two groups of dots move at constant velocity with opening angle γ. **e** The direction in which human perception of the opening angle is biased depends on the true opening angle. Black dots: human data, reproduced from ref. 36, error bars denote S.E. of the mean across subjects; *n* = 3 subjects, 80 trials per angle and subject. Purple line: model percept. Insets: the model's inferred motion decomposition. **f** Varying the contrast of one dot group modulates the biased percept of the angle of the other group. Purple: model percept for γ = 45°, qualitatively matching data from ref. 37. Blue: predicted inversion of the bias for smaller opening angles. **g** Same as panel **f**, but for varying the speed of the second group. Purple: model percept for γ = 60°, qualitatively matching data from ref. 38. Dashed blue: model percept for γ = 90°, qualitatively matching data from ref. 36. Solid blue: predicted biphasic function for smaller opening angles. **h–l** Extended experiment from ref. 39 which surrounds the two central RDKs with additional RDKs in an annulus. The hierarchical inference model replicates human perception in various conditions. **h** A surround with dots moving vertically both up- and downwards ("bi-directional surround" in ref. 39, indicated by orange arrows in the top-left sketch's annulus) causes no repulsion in the perceived directions of horizontally moving RDKs in the center (darker orange arrows in the top-left sketch's center). Our model replicates this perception as shown in the histogram of 200 trial repetitions. **i** Coherently moving annulus RDKs cause the perceived inner velocities to be biased away from the surround direction. **j** For diagonally moving inner RDKs, the same coherent downward surround has no noticeable effect. **k** Neither does a bi-directional surround bias the percept of diagonally moving inner RDKs. **l** An upward surround, in contrast, biases the percept of the inner RDKs to close-to-horizontal motion. Source data are provided as a Source Data file.

additional RDKs in the annulus move up and down, respectively. For this stimulus human observers show no direction repulsion[39]. We simulated this extended MDR experiment with our hierarchical inference model by extending the motion tree of Fig. 4b to include two group components for the outer and inner RDKs, respectively, on the third level, and four individual components (one per RDK) as leaves, on the forth level (cf. Supplementary Fig. 5). Across 200 simulated trials (see Methods), the distribution of inner RDK directions perceived by the model at trial end (see histogram in Fig. 4h) match the reported unbiased perception of humans.

Our model was further able to replicate human perceptual biases for various other combinations of dot motion in the inner and surrounding RDKs explored by Takemura et al. (see Fig. 4i–l, and Supplementary Fig. 5 for example trials). The percepts to all combinations are qualitatively replicated by our model. When both surrounding RDKs move downward, as shown in Fig. 4i, the perceived motion of the inner RDKs is slightly biased upward. The reason for the bias in the model's percept is a small illusory self-motion component in upward direction which necessitates a slight diagonal upward tilt of the inner RDKs' individual motions for explaining their horizontal retinal velocities. When modifying the stimulus such that the inner RDKs move diagonally with a 90 degree opening angle (see Fig. 4j–l), human and model percepts remain unbiased in the case of downward (Fig. 4j) and

bi-directional surrounding motion (Fig. 4k). In both cases, the directional contrast of the presented velocities obviates the illusory identification of self-motion, thereby implicating unbiased percepts of the model. If, however, the surrounding RDKs move upwards, strong direction repulsion on the inner dots was reported[39] leading to their perceived motion to become almost horizontal (Fig. 4l). In the model, this effect originates from illusory downward self-motion arising from the general alignment of the presented velocities. Overall, our hierarchical inference model replicated biased and unbiased perception across a variety of stimulus conditions.

Turning to noise-dependent motion integration of spatially distributed stimuli, we investigated a motion illusion by Lorenceau[42] which has received little attention in the literature (see Fig. 5). Two groups of dots oscillate in vertical and horizontal orientation, respectively (see Fig. 5a and Supplementary Movie 3). Both groups follow sine-waves with identical amplitude and frequency, but maintain a relative phase shift of π/2 that is consistent with an imaginary global clockwise (CW) rotation (indicated by a gray arrow in Fig. 5a). This stimulus can be considered to be location-indexed, as the small oscillation amplitude of less than 1 degree of visual angle caused the stimulus to conveniently fit into the receptive fields of individual neurons of the human homolog of area MT[47]. Interestingly, the stimulus' percept changes once disturbances orthogonal to the axes of

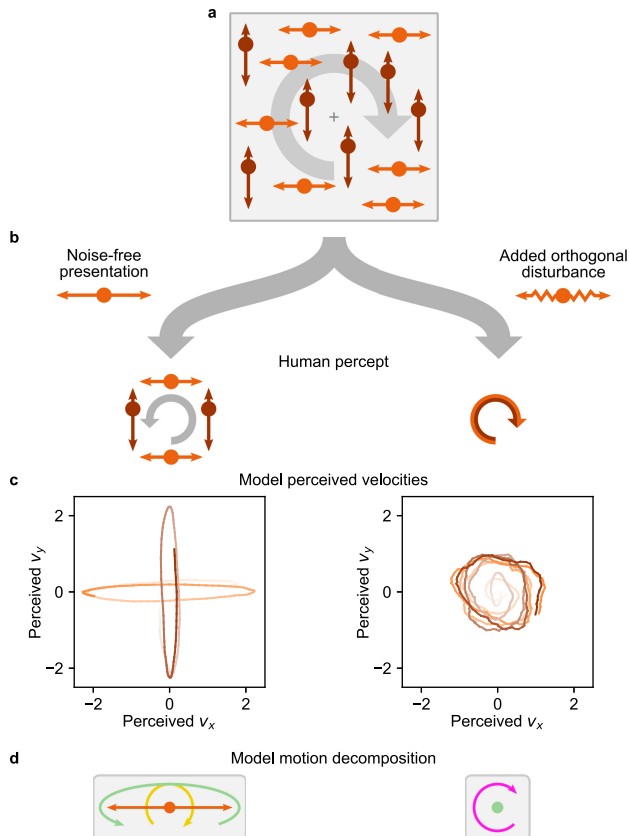

**Fig. 5 | Noise-dependent perceptual changes for motion integration of spatially distributed stimuli. a** In the motion illusion from Lorenceau[42], a vertically and a horizontally oscillating group of dots maintain a 90°-phase shift consistent with global clockwise rotation (indicated as gray arrow). **b** The noise-free stimulus (left branch) evokes transparent motion with an additional counter-clockwise rotating percept in human observers. Adding motion noise by disturbing dot trajectories orthogonally to their group's oscillation axis (right branch; modeled by increased observation noise $\sigma^2_{obs}$) flips the percept to a single coherent rotation of all dots in clockwise direction. **c** The model's perceived velocities in both stimulus conditions (time = color gradient from low to high contrast; $t \le 2$ s in noise-free condition; $t \le 5$ s in noisy condition). For visual clarity, perceived velocities have been smoothed with a 200 ms box filter for plotting. **d** Illustration of the model's inferred motion decomposition. For noise-free stimuli, clockwise rotating self-motion is compensated by counter-clockwise rotating group motion (sketched here for the horizontal group). With motion noise, only a single, clockwise rotating shared motion component is inferred for all dots. Source data are provided as a Source Data file.

oscillation are added (called "motion noise" in ref. [42], see Fig. 5b). Without motion noise, participants perceive transparent motion, that is, the dots within either group are combined to a rigidly moving object according to common fate, and both groups are perceived as moving separately. Their movement, however, is not perceived as strictly vertically and horizontally, but rather the stimulus induces an impression of slight counter-clockwise (CCW) rotation, that is, "opposite to veridical"[42]. With motion noise, in contrast, the percept switches in two ways: all dots appear to move coherently along a circle, and the perceived direction of movement becomes CW. These percepts are illustrated in Fig. 5b.

Applied to this stimulus, our model replicates the perceived rotation direction reversal with increased motion noise, which we simulated through an increase in the observation noise $\sigma^2_{obs}$. Specifically, the model's perceived velocities for both groups of dots featured a slight global CCW rotation on top of two generally separated groups for the noise-free stimulus, and a single global CW rotation once observation noise is increased (Fig. 5c). Inspecting the model's motion

decomposition provides a possible answer to how this flip in perceived rotation emerges, which is illustrated in Fig. 5d by the example of the horizontal group. On noise-free presentation, dot motion was decomposed into clockwise rotating self-motion (golden arrow) plus a horizontally elongated, yet slightly CCW rotating group motion (green arrow), leading to the transparent CCW motion percept. Once observation noise increased, the inferred motion structure discarded the separated groups in favor of a single global motion component (magenta), leading to the percept of coherent CW rotation for all dots (see Supplementary Fig. 6 for trajectories of the motion strengths and sources under both conditions).

## Object recognition and perceptual switching of nested structure-from-motion displays

Motion relations do not only aid dynamic tasks, such as tracking and prediction, but also provide essential cues for object recognition. Structure-from-motion (SfM), the perception of 3D objects from 2D visual displays, is well-studied in psychology[48–54] and neuroscience[55–58]. We asked whether our model can support SfM perception and replicate the salient phenomenon of perceptual switching when presented with ambiguous stimuli (see Fig. 6a). Furthermore, using the model, we identified SfM displays of nested objects which could inspire future psychophysics experiments studying how structure interacts with perceptual ambiguity.

Typical SfM displays, like the point cloud-cylinder in Fig. 6a and Supplementary Movie 4, involve rotational motion in three dimensions, contrasting with the translational motion in two dimensions considered so far. Our generative model supports such 3D rotation in location-indexed experiments: as illustrated in Fig. 6b, introducing a rotational motion source, $s^{rot}$, which describes the cylinder's angular velocity around the y-axis, yields a linear dependence of the observed retinal velocities on $s^{rot}$ at every input location (dashed orange circles) owing to the locations' fixed coordinates. Thus, rotational motion is supported naturally by the component matrix, $C$, (cf. Fig. 1e) and integrates without any changes into our hierarchical inference model.

Ambiguous SfM displays, such as the considered frontal view of a rotating cylinder, furthermore feature equivocal correspondences of spatially overlapping inputs to the cylinder's surface at the front and back. Mentally assigning the overlapping left- and rightward retinal velocities to their depth locations is key to forming a coherent percept of the 3D object. To support such percepts in our model, we added a basic assignment process: spatially overlapping velocities are assigned to their depth location on the cylinder (front or back) such that the assignment locally minimizes the model's prediction error, $\epsilon_t$, in Eq. (2). Furthermore, this assignment is independently re-evaluated at each input location with a uniform probability in time (see Methods). We tested the model's ability to perceive SfM by using a motion tree with self-motion, rotational motion and individual motion (see Fig. 6c, and Supplementary Fig. 7 for a control simulation with more motion components). As shown in Fig. 6d, the model swiftly identifies rotational motion across all input locations at a constant angular speed, matching the human percept of a rotating cylinder. Subsequently, the percept switches randomly and abruptly between CW and CCW rotation, with inter-switch-intervals following a Gamma distribution (see Fig. 6e). The resulting stochastically switching percepts with typical durations of a few seconds match the reported bistable perception of humans[53,57].

To explore how more complex structures could interact with SfM perception, we asked how our model interprets the rotation of nested point-cloud cylinders (see Fig. 6f and Supplementary Movie 4). Their rotation is easily identified by humans[49], and features a more complex structure than basic SfM displays that only require a single rotational motion source. To present this stimulus to the model, the extended graph in Fig. 6g features rotational sources not only for the inner and outer cylinders (light and dark

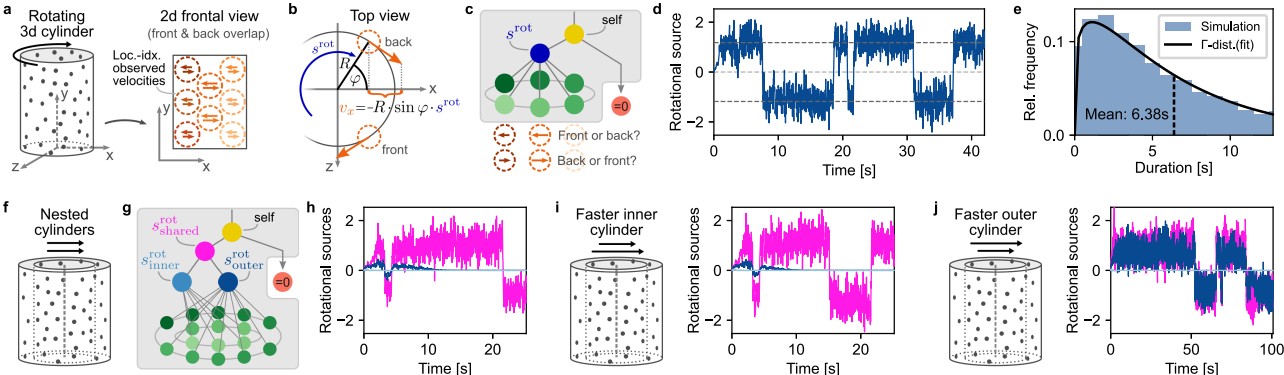

**Fig. 6 | Object recognition and perceptual switching of nested structure-from-motion (SfM) displays. a** Cylindrical SfM stimulus. A random point cloud on the surface of a rotating, transparent cylinder (left) supports two possible percepts when viewed from the front without depth information (right). Humans perceive the structured motion of this 2D projection as a rotating 3D cylinder, albeit with bistable direction of the perceived rotation. **b** Top view illustration of how the generative model supports rotational motion. The rotational motion source, $s_t^{rot}$, describes angular velocity about the vertical axis ($s^{rot} > 0$ for CCW rotation, by definition). In location-indexed experiments, observed velocities, $v_t$, at a (fixed) location with angle $\varphi$ and radius $R$ are a linear function of the rotational motion source. In the frontal view of SfM experiments, only the x-component, $v_x = -R\sin(\varphi) s_t^{rot}$, and the vertical y-component, $v_y = 0$, are visible. **c** Motion tree and correspondence problem. The graph contains self-motion, rotational motion of the entire cylinder, and individual motion for every location. For any x-y coordinate, there exist two overlapping observed velocities which are ambiguous regarding their depth position (front or back). We performed the assignment of observations to their perceived depth (front or back) such that the prediction error, $\epsilon_t$, in Eq. (2) is minimized. **d** 3D percept and perceptual bistability. Like humans, the model identifies rotation as the single motion component. The value of $s_t^{rot}$

switches randomly between CW and CCW rotation with constant angular speed. **e** Distribution of perceptual switches. The distribution of duration-of-percepts closely follows a Gamma distribution, as commonly reported in human psychophysics. **f** Extension of the SfM display adding a smaller point cloud-cylinder, nested within the original cylinder. **g** Motion tree for the extended experiment. Three rotational components are provided: shared rotation of both cylinders, rotation of the outer cylinder, and rotation of the inner cylinder. The correspondence problem now demands assigning 4 observations where both cylinders overlap. **h** Perceived structure for identical angular speed of both cylinders. The model infers a single shared rotational component. **i** Fast inner cylinder. When increasing the angular speed of the inner cylinder by 50% (sketch on the left), the inferred structure is unaffected (right): the cylinders are perceived as having the same angular velocity. **j** Fast outer cylinder. In contrast, when increasing the angular speed of the outer cylinder by 50% (left), the cylinders' speeds are perceived as separated (right). For visual clarity, the trees in panels c and g show only 5 and 3 receptive field locations for the outer and inner cylinder, respectively, while for the simulations, we used 7 and 5 locations. Source data are provided as a Source Data file.

blue, respectively), but also the possibility of shared motion (magenta) affecting both cylinders. Where both cylinders overlapped, the assignment now minimized the prediction error over 4 overlapping retinal velocities (24 possible combinations per location), but remained otherwise unchanged. When both cylinders rotated with the same angular velocity of 90°/s, the model inferred a single shared rotational component (see Fig. 6h) leading to the impression of rigid rotation in which perceptual switches occur simultaneously for both cylinders. Identifying a structure with a single component rather than separate rotations for both cylinders is the result of the model's preference of simple structures. Increasing the angular velocity of the inner cylinder by 50% to 135 °/s (see Fig. 6i) did not change the model's percept of a rigidly shared rotation, but led to a slightly higher perceived speed of rotation. Inspecting the inference process revealed that the assignment process often assigned fast-moving dots of the inner cylinder to the outer cylinder and, vice versa, slower moving outer dots to the inner cylinder. This assignment yielded a sufficiently coherent interpretation of all retinal velocities as originating from a single rotation (within the bounds of perceptual acuity, $\sigma_{obs}$) for the model to prefer the simpler structure. Finally, a display in which the outer cylinder rotates faster than the inner cylinder (135 °/s and 90 °/s, respectively; see Fig. 6j) changed the model's inferred structure to perceiving different rotational speeds for both cylinders. Yet, even though each cylinder had its distinct perceived rotation, their rotational directions remained aligned and perceptual switches still occurred simultaneously, a perceptual linkage known from related experiments[54].

The nested SfM displays in Fig. 6f–j provide testable predictions for future psychophysics studies (see Supplementary Movie 4 for a video of all conditions). The model's percepts across all conditions matched the percept of the authors.

## Experimental predictions from a biological network model of hierarchical inference

Finally, we asked whether and how a biologically plausible neural network could implement our online hierarchical inference model. To this end, we devised a recurrent neural network model of rate-based neurons. Naturally, such modeling attempt relies on many assumptions. Nonetheless, we were able to identify several experimentally testable predictions that could help guide future neuroscientific experiments.

Following Beck et al.[59], we assumed that task-relevant variables can be decoded linearly from neural activity ("linear population code") to support brain-internal readouts for further processing, actions and decision making. Furthermore, we employed a standard model for the dynamics of firing rates, $r_i(t)$, and assumed that neurons can perform linear and quadratic integration[59–62]:

$$\tau_i \, \partial_t r_i = -r_i + f_i(\boldsymbol{w}_i^\top \boldsymbol{r} + \boldsymbol{r}^\top \boldsymbol{Q}^{(i)} \boldsymbol{r}), \tag{4}$$

with time constant $\tau_i$, activation function $f_i(\cdot)$, weight vector $\boldsymbol{w}_i$ and matrix $\boldsymbol{Q}^{(i)}$ for linear and quadratic integration, respectively. The rate vector, $\boldsymbol{r}(t)$, here comprises all presynaptic firing rates, including both input and recurrent populations. With these assumptions, we derived a network model with the architecture shown in Fig. 7a, which implements the online model, given by Eq. (1)–(3), via its recurrent interactions and supports linear readout of all task-relevant variables. That is, for every task-relevant variable, $x$, there exists a vector, $\boldsymbol{a}_x$, such that $x = \boldsymbol{a}_x^\top \boldsymbol{r}$ (see Supplementary Note 4 for the derivation).

The network consists of three populations. The input population (bottom in Fig. 7a) encodes the observed velocities, $\boldsymbol{v}_t/\sigma_{obs}^2$, and observation precision, $1/\sigma_{obs}^2$, in a distributed code. While any code that supports linear readout of these variables could serve as valid neural input, we chose a specific model that, as shown below,

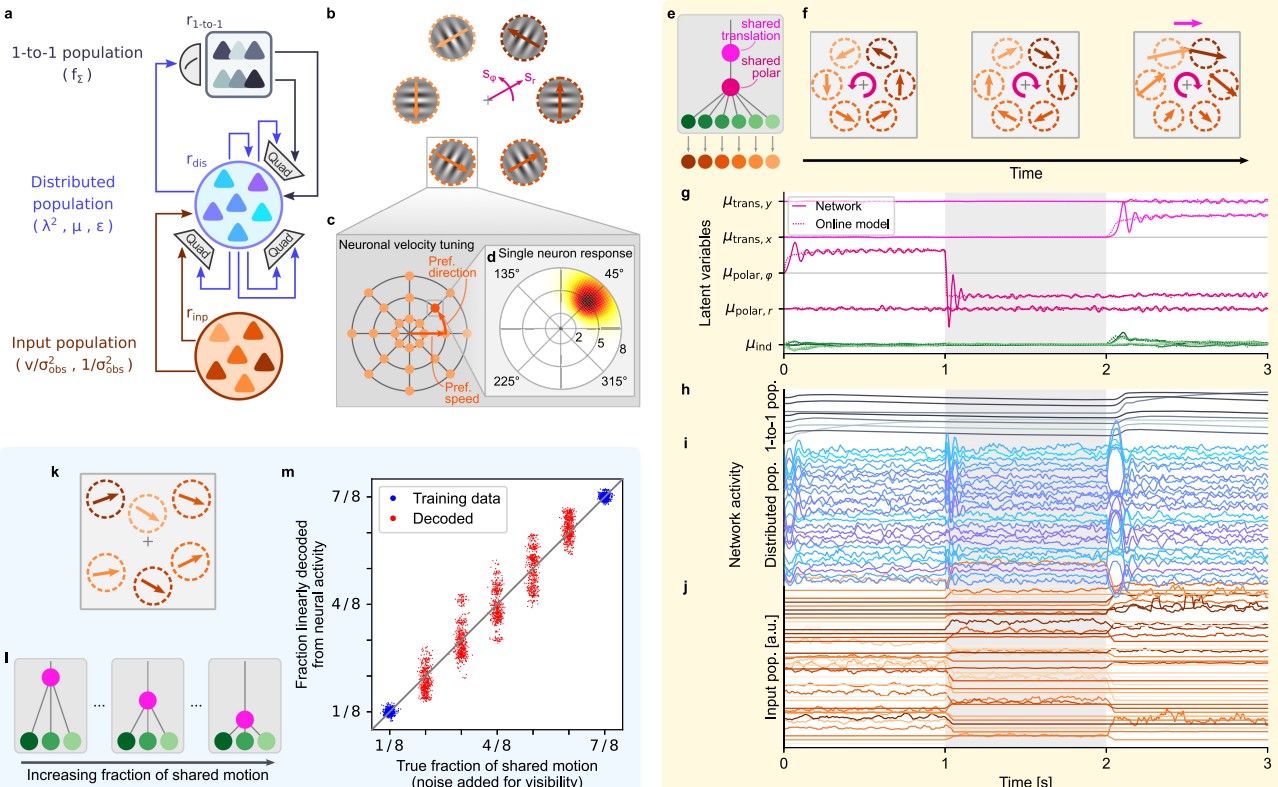

**Fig. 7 | Hierarchical inference can be performed by a biologically realistic network model.** **a** Network model implementing the online hierarchical inference model. Linear and quadratic interactions are indicated by direct arrows and Quad boxes, respectively. In parentheses, the variables represented by each population. **b** Rotational stimulus in a location-indexed experiment. Besides translational (Cartesian) motion, the model also supports rotational, $s_\varphi$, and radial motion, $s_r$. **c** Tuning centers in a model of area MT. A local population of neurons, which share the spatial receptive field highlighted in panel **b**, cover all directions and speeds with their velocity tuning centers. **d** Response function for the neuron highlighted in panel **c**. The neuron responds strongly to local velocities into the upper-right direction with a speed of ~5°/sec. Max. rate = 29.5 spikes/s. **e** Motion structure used for the network simulation in panels **f**–**j**, including simultaneous translational, rotational and radial motion sources. **f** Illustration of the stimulus. After 1 s of counter-clockwise rotation around the fixation cross, the rotation switches to clockwise. At $t = 2$ s, rightward translation is superimposed on the rotation. **g** Motion sources inferred by the network (solid lines: distributed population read-out; dotted lines: solution by the online model given by Eqs. (1)–(3)). Shown is $\mu_t$ for

translational, rotational, radial and individual motion. Only 4 individual components (2 x- and 2 y-directions) are shown for visual clarity. **h** Firing rates of the 1-to-1 population. Rates are in arbitrary units (a.u.) because the theory supports scaling of firing rates with arbitrary factors. **i** Same as panel **h**, but for a random subset of 25 neurons of the distributed population. **j** Same as panel **h**, but for a random subset of 40 neurons of the input population, and smoothed with a 50 ms box filter for plotting. **k** Stimulus of a proposed neuroscience experiment. Velocities in distributed apertures follow the generative model from Fig. 1 using shared motion and individual motion. **l** Different trials feature different relative strengths of shared and individual motion, ranging from close-to-independent motion (left) to highly correlated motion (right). **m** Linear readout of the fraction of shared motion from neural activity. Seven different fractions of shared motion were presented (x-axis; noise in x-direction added for plotting, only). A linear regression model was trained on the outermost conditions (blue dots). Intermediate conditions were decoded from the network using the trained readout (red dots). Only a subset of $7 \times 500 = 3500$ points is shown for visual clarity. Source data are provided as a Source Data file.

captures many properties of motion-sensitive area MT. The distributed population (center in Fig. 7a) simultaneously represents the squared motion strengths, $\lambda_t^2$, mean of the sources, $\mu_t$, and prediction errors, $\epsilon_t$, in a distributed code with linear readout. For those, almost arbitrary readouts suffice, such that we chose randomly generated readout vectors, $a$. Notably, we propose the prediction errors, $\epsilon_t$, to be linearly decodable, which allowed Eq. (2) to be implemented with the neuron model in Eq. (4) (see Supplementary Note 4, Sections 3 and 4). All neurons in the distributed population have simple activation functions, $f_i(\cdot)$, that are linear around some baseline activity. The linear decodability of $\lambda_t^2$, $\mu_t$, and $\epsilon_t$ are testable predictions. Finally, the 1-to-1 population (top in Fig. 7a) represents the uncertainty, $\Sigma = f_\Sigma(\lambda^2)$, in a one-to-one mapping, $r_m \propto f_\Sigma(\lambda_m^2)$, with $r_m$ being the firing rate of either a single cell or, more likely, a small population. The theoretical motivation behind this representation is two-fold: on the one hand, the non-linear form of $f_\Sigma(\cdot)$ prevents a distributed, linearly decodable representation (see Supplementary Note 4, Section 5); on the other hand, the particular shape of $f_\Sigma(\lambda_m^2)$, shown in Fig. 1i, mirrors the typical activation function of Type-I

neurons[63], such that the proposed representation emerges naturally for the activation function, $f_\Sigma(a_{\lambda_m^2}^\top r)$, in the 1-to-1 population (using the fact that $\lambda_m^2$ can be read out neurally with weights $w = a_{\lambda_m^2}$). Overall, the network structure predicts $\lambda_t^2$, $\mu_t$, and $\epsilon_t$ to be linearly decodable, and the components of $f_\Sigma$ to be independently encoded in single neurons or small neural populations.

Even though the network model supports both the object-indexed and location-indexed experiments from Figs. 2–6, the retinotopic organization of the early visual system[21,64] brings a location-indexed perspective closer in line with our understanding of how the cortex encodes visual information. Furthermore, as we show in Supplementary Note 1, Section 5, our model can be extended to support motion sources in polar coordinates (see Fig. 7b), such that it supports salient real-world retinal input motifs, such as rotation and radial expansion/contraction about the fovea. (Note that rotation and expansion on the retina are conceptually distinct from the cylindrical rotation, $s^{rot}$, in structure-from-motion, discussed earlier.) Representations of angular motion, $s_\varphi$, and radial motion, $s_r$, can also coexist with translational motion (i.e., linear motion in Cartesian coordinates)

within the same population. Selective neural response to rotation, expansion/contraction and translation, as well as combinations thereof, such as spiraling, has been frequently reported in the dorsal medial superior temporal area (MSTd)[19,65].

Before demonstrating this capability in simulations, let us provide further information about the model's input population, and how it relates to known properties of area MT. To do so, consider the location-indexed stimulus in Fig. 7b. During fixation, each aperture stimulates a population in retinotopically organized, motion sensitive area MT[21]. Neurons in MT are tuned to respond preferentially to a certain direction and speed (Fig. 7c), such that the full population jointly covers all velocities in a polar grid[66,67]. The response of individual neurons to velocities within their spatial receptive field is commonly modeled by a log-normal function for speed[67] and a von Mises function for direction[68], leading to the bump-like response function shown in Fig. 7d. As a third factor, higher visual contrast (smaller $\sigma_{\text{obs}}^2$) leads to higher firing rates[69]. As we derive in Supplementary Note 4, Section 6, a neural population with these response functions supports linear readout of input velocities, $\boldsymbol{v}_t/\sigma_{\text{obs}}^2$, and precision, $1/\sigma_{\text{obs}}^2$, in Cartesian coordinates. This provided us with a biologically realistic and, at the same time, theoretically grounded input population model which we used in the following network simulations.

We tested the network's ability to perform online hierarchical inference in the simulation shown in Fig. 7e–j. To challenge the network, we employed a stimulus that combined shared rotation and shared translation (motion tree in Fig. 7e). Six input populations with receptive fields shown in Fig. 7f projected to a distributed population of 100 neurons and a 1-to-1 population of size 8 (one per motion strength). After one second of retinal velocities of counter-clockwise rotation (Fig. 7f, left), these velocities switched to clockwise rotation (center), followed by a superposition of clockwise rotation and rightward translation (right). As the network response for the three populations to this stimulus shows (Fig. 7h–j), input neurons fired sparsely and were only active if the stimulus matched their preferred direction or speed. Neurons in the distributed population, in contrast, showed fluctuating activity with little apparent structure, and exhibited population-wide transients upon changes of the input. Finally, the 1-to-1 population responded more graded and with a short delay, suggesting that every rate, $r_m$, describes a small cortical population rather than individual neurons. Knowledge of the (randomly drawn) vectors, $a_x$, of the simulated network, allowed us to read out the network's latent motion decomposition at each time point (solid lines in Fig. 7g). This revealed that the network correctly decomposed the input, including the overlaid rotational and translational motion, and closely matched the online model (dotted lines).

In experiments with humans and animals, we have no access to these readout vectors, $\boldsymbol{a}_x$. We therefore simulated a possible experiment that tests our model and doesn't require this knowledge (see Fig. 7k–m), while benefiting from precise stimulus control. Several apertures, located at the receptive fields of recorded neurons in motion sensitive areas (e.g., area MT or MSTd), present a motion stimulus according to the generative model from Fig. 1. Velocities across the apertures are positively correlated owing to a shared motion source, but also maintain some individual motion (see Fig. 7k and Supplementary Movie 5). A series of trials varies the fraction of shared motion in the stimulus, $q := \lambda_{\text{shared}}^2/(\lambda_{\text{shared}}^2 + \lambda_{\text{ind}}^2)$, ranging from almost independent motion (Fig. 7l, left) to almost perfect correlation (right). According to the network model, $\boldsymbol{\lambda}^2$ can be read out linearly. For the simulation in Fig. 7m, we presented the network with trials of seven values of $q$. We then trained a linear regression model to predict $q$ from the neural activity for the two most extreme structures (blue dots in Fig. 7m), and decoded $q$ for the intermediate structures using this regression model (red dots in Fig. 7m). Owing to the stochastic stimulus generation, the network's motion structure estimates, $\boldsymbol{\lambda}_t$, fluctuate around the true strength—yet, on average, the trained linear

readout correctly identified the fraction of global motion in the stimulus. This is a strong prediction of the network model, which could be tested in a targeted neuroscientific experiment.

## Discussion

We have proposed a comprehensive theory of online hierarchical inference for structured visual motion perception. The derived continuous-time model decomposes an incoming stream of retinal velocities into latent motion components which in turn are organized in a nested, tree-like structure. A scene's inferred structure provides the visual system with a temporally robust scaffold to organize its percepts and to resolve momentary ambiguities in the input stream. Applying the theory to human visual motion perception, we replicated diverse phenomena from psychophysics in both object-indexed and location-indexed experiment designs. Furthermore, inspection of the model's internal variables provided normative explanations for putative origins of human percepts and spawned concrete predictions for psychophysics experiments. Finally, the online inference model afforded a recurrent neural network model with visual inputs reminiscent of cortical area MT and latent structure representations reminiscent of area MSTd.

Our online model shares features with predictive coding[70,71], a theory positing that "higher" brain areas provide expectations to earlier areas in a hierarchical model of sensory input and that neural processing aims to minimize prediction errors between top-down expectations and bottom-up observations. Like predictive coding, the dynamics in Eq. (2) update the values of motion sources to minimize prediction errors, $\boldsymbol{\epsilon}_t$, within the bounds imposed by the identified structure. Yet, structure identification according to Eq. (1) follows a different principle by computing a running average of motion source magnitudes. This contrasts with common theories of predictive coding in the brain[72,73], which assume that the same computational principle is repeated across cortical hierarchies, and demonstrates how hierarchical visual processing could combine multiple interacting algorithmic motifs. Moreover, the network model in Fig. 7a challenges the prevalent view[72,74] that error signals are necessarily represented by distinct neural populations (or alternatively distinct dendritic compartments[75]). While our network model supports the possibility of distinct error populations, we show that prediction errors could also be computed and conveyed by the same neurons representing other quantities, such as the motion sources, $\boldsymbol{\mu}_t$, and even the structure, $\boldsymbol{\lambda}_t^2$, using a distributed neural code.

In the main text, we have for the sake of clarity limited the presentation of the theory to a basic version that nonetheless covers all essential concepts. In Supplementary Note 3, we present several extensions that are naturally covered by our model:

(i)   observation noise, $\sigma_{\text{obs}}$, can be time- and object-dependent, which is relevant for modeling temporary occlusion of a subset of stimuli;

(ii)  observation noise can be non-isotropic (different values in x- and y-direction), which is relevant for angle-dependent edge velocities in apertures[76];

(iii) for optimal inference, different motion components can feature different time constants, since velocity is expected to change more slowly for heavy objects due to higher inertia;

(iv)  different motion components may tend to co-occur or exclude one another in real-world scenes, which can be modeled by an interaction prior of pairwise component compatibility; and

(v)   when motion components are not present for a long time, they will decay to zero, preventing their rediscovery, which can be mitigated by a prior on motion strengths.

The current theory is limited to velocities as input, thereby ignoring the well-documented influence of spatial arrangement on visual motion perception, such as center-surround modulation[77,78],

adjacency[26] or motion assimilation[79], as well as Gestalt properties[80]. Furthermore, the model does not solve the correspondence problem in object-indexed experiments, but simply assumes that velocities are correctly assigned to the input vector as objects move about the visual field. For location-indexed experiments, we have explored how structure inference in concert with a basic assignment process, which minimizes the observer's local prediction errors, could solve the correspondence problem during structure-from-motion perception. Our work focuses on the simultaneous inference of motion sources, $s_t$, and motion strengths, $\lambda_t$. Other quantities, such as time constants and, probably more importantly, the motion components, $C$, have been assumed to be given. It is worth noting, however, that gradient-based learning of $C$ is, in principle, supported by the theory on long time scales (see Supplementary Note 3, Section 5). Finally, limited experimental evidence of the neural correlates of motion structure perception required the neural network model to rely on many modeling assumptions. The model's predictions should act as a starting point for further scientific inquiry of these neural correlates.

Even though the sensory processes underlying object-indexed motion perception necessarily differ from those of location-indexed perception, our model describes human perception for both types of experiments. Thus, both types might share the same underlying neural mechanisms for structure inference. This raises the intriguing question whether there exist stable, object-bound neural representations of velocity. Furthermore, our work points towards a tight link between neural representations of latent structure and representations of uncertainty in that the estimated motion strengths, $\lambda_t$, determine the credit assignment of prediction errors through the gating function, $f_\Sigma(\lambda_t^2)$—a function that also computes the variance of motion components, e.g., the brain's uncertainty about flock velocity. Behaviorally, sensory noise directly impacts the perceived structure of a scene as demonstrated experimentally by the perceptual reversal in the Lorenceau-motion illusion[42] (cf. Fig. 5). More generally, our theory predicts that the visual system will organize its percepts into simpler structures when sensory reliability decreases. Moreover, the reliability of visual cues plays a role in multisensory integration[81], with area MSTd[82,83], but not area MT[84], exhibiting tuning to vestibular signals. Thus, MSTd may be a candidate area for multisensory motion structure inference. Overall, we expect our theoretical results to guide targeted experiments in order to understand structured visual motion perception under a normative account of statistical information processing.

## Methods

In what follows, we provide an overview of the generative model, the online hierarchical inference model, the computer simulations, and the data analysis. A more detailed presentation is found in the Supplementary Information.

### Generative model of structured motion

We consider $K$ observable velocities, $v_{k,d}(t)$, in $D$ spatial dimensions. For notational clarity, we will consider in this Methods section only the case $D = 1$ and use the vector notation, $\boldsymbol{v}_t = (v_1(t),..,v_K(t))^\top$. The extension to $D > 1$ is covered in Supplementary Note 1, Section 4. Observable velocities, $\boldsymbol{v}_t$, are generated by $M$ latent motion sources, $s_{m,d}(t)$, abbreviated (for $D = 1$) by the vector $\boldsymbol{s}_t = (s_1(t),..,s_M(t))^\top$. Velocities are noisy instantiations of their combined ancestral motion sources, $\boldsymbol{v}_t \sim \mathcal{N}(\boldsymbol{C}\boldsymbol{s}_t, \sigma_{\mathrm{obs}}^2/\delta t\, \boldsymbol{I})$, where $C_{km} = +1, -1$, and 0 in $K \times M$ component matrix, $C$, denote positive, negative and absent influence, respectively. For the formal definition, observations, $\boldsymbol{v}_t$, remain stable within a short time interval $[t, t + \delta t]$, and the observation noise variance, $\sigma_{\mathrm{obs}}^2/\delta t$, ensures a $\delta t$-independent information content of the input stream. In the online inference model,

below, we will draw the continuous-time limit, which will become independent of $\delta t$. In computer simulations, $\delta t$ is the inverse frame rate of the motion display (default value: $1/\delta t = 60$ Hz). Each motion source (in each spatial dimension) follows an Ornstein–Uhlenbeck process, $\mathrm{d}s_m = -s_m/\tau_s\, \mathrm{d}t + \lambda_m\, \mathrm{d}W_m$, with time constant $\tau_s$, motion strength $\lambda_m$ (shared across dimensions), and Wiener process $W_m$. The OU process's equilibrium distribution, $\mathcal{N}\left(0, \frac{\tau_s}{2}\lambda_m^2\right)$, introduces a slow-velocity prior which, as we note, has recently been proposed to originate from the speed-contrast statistics of natural images[85]. The resulting marginal stationary velocity distribution of $v_k$ is $v_k \sim \mathcal{N}\left(0, \sigma_{\mathrm{obs}}^2/\delta t + \frac{\tau_s}{2}\sum_{m=1}^{M} C_{km}^2 \lambda_m^2\right)$.

**Radial and rotational motion sources.** In location-indexed experiments, the input's location (e.g., a neuron's receptive field) remains fixed. For $D = 2$, the fixed input locations enable our model to support rotations and expansions around various axes. In this manuscript, we consider two cases: rotation around a vertical axis (SfM experiment in Fig. 6) and rotation/expansion around the fovea (network model in Fig. 7).

For rotations around a vertical axis, each input $\boldsymbol{v}_k$ has fixed polar coordinates $(R_k, \varphi_k)$ as sketched in Fig. 6b. When describing rotation by means of a rotational motion source, $s_t^{\mathrm{rot}}$, we obtain for the noise-free part of the observed velocity in Cartesian coordinates: $v_{k,x} = -R_k \sin(\varphi_k) s_t^{\mathrm{rot}}$, $v_{k,y} = 0$, and $v_{k,z} = -R_k \cos(\varphi_k) s_t^{\mathrm{rot}}$. Owing to the linear dependence of $\boldsymbol{v}_k$ on $s^{\mathrm{rot}}$, we can include the coefficients as a column in component matrix, $C$, and $s^{\mathrm{rot}}$ as a motion source in the vector $\boldsymbol{s}_t$. Note that in the SfM experiments only the x- and y-directions are observed.

Similarly, for rotation/expansion around the fovea, each input $\boldsymbol{v}_k$ has fixed polar coordinates $(R_k, \vartheta_k)$ with radial distance $R_k$ and angle $\vartheta_k$, relative to the pivot point (we use different symbols than for vertical rotation for notational clarity). Denoting radial and rotational motion sources by $s_r$ and $s_\varphi$, we obtain for the noise-free part of $\boldsymbol{v}_k$ in Cartesian coordinates: $v_{k,x} = s_r \cos \vartheta_k - s_\varphi R_k \sin \vartheta_k$, and $v_{k,y} = s_r \sin \vartheta_k + s_\varphi R_k, \cos \vartheta_k$. Since $R_k$ and $\vartheta_k$ are fixed coefficients, the mapping $(s_r, s_\varphi) \mapsto (v_{k,x}, v_{k,y})$ is linear and, thus, can be described by the component matrix $C$. The full derivation and an illustration of the velocity relations in polar coordinates are provided in Supplementary Note 1, Section 5.

### Online inference

The goal of motion structure inference is to simultaneously infer the value of motion sources, $s_t$, and the underlying structure, $\lambda$, from a stream of velocity observations. The number of spatial dimensions, $D$, component matrix, $C$, time constant $\tau_s$, and observation noise $\sigma_{\mathrm{obs}}$ are assumed to be known. The EM algorithm leverages that changes in $s_t$ and $\lambda$ (if changing at all) occur on different time scales, $\tau_s$ and $\tau_\lambda$, respectively. For $\tau_\lambda \gg \tau_s$, the EM algorithm treats $\lambda$ as a constant for inferring $s_t$ (E-step), and optimizes an estimate, $\lambda_t$, online based on the inferred motion sources (M-step).

**E-Step.** For fixed $\lambda$, the posterior $p(\boldsymbol{s}_t | \boldsymbol{v}_{0:t}; \lambda)$ is always a multivariate normal distribution, $\mathcal{N}(\boldsymbol{\mu}_t, \boldsymbol{\Sigma}_t)$, and can be calculated by a Kalman-Bucy filter[86,87]; see Supplementary Note 2, Sections 1, 2, and 3.1 for the derivation. This yields coupled differential equations for the time evolution of $\boldsymbol{\mu}_t$ and $\boldsymbol{\Sigma}_t$. To reduce the computational complexity of the system, we perform an adiabatic approximation on the posterior covariance, $\boldsymbol{\Sigma}_t$, by assuming (a) that it has always converged to its stationary value, and (b) that off-diagonal values in $\boldsymbol{\Sigma}_t$ are zero, that is, we ignore correlations in uncertainty about latent motion sources in the posterior distribution. As shown in the full derivation in Supplementary Note 2, Section 3, the first assumption is warranted because the stationary value of $\boldsymbol{\Sigma}_t$ depends only on the current structure estimate, $\lambda_t$; then, because $\boldsymbol{\Sigma}_t$ decays to stationarity at time scale $\tau_s/2$, it can always follow any changes in $\lambda_t$ which happen at time scale $\tau_\lambda \gg \tau_s$. The

**Table 1 | Default parameters of the computer simulations**

| Description | Variable | Object-indexed | Location-indexed | Network |
|---|---|---|---|---|
| Time const. motion sources | $\tau_s$ | 0.300 s | 0.100 s | 0.100 s |
| Time const. motion strengths | $\tau_\lambda$ | 1.000 s | 0.333 s | 0.333 s |
| Inv. observation frame rate | $\delta t$ | 1/60 s | 1/60 s | 1/120 s |
| Observation noise | $\sigma_{\text{obs}}$ | 0.05 | 0.017 = 0.05 ÷ 3 | 0.017 |
| Initial motion strength | $\lambda_m(t=0)$ | 0.5 | 0.5 | 0.5 |
| No. of pseudo observation | $\nu_m$ | 0 | 0/–1 | 0 |
| Val. of pseudo observations | $\kappa_m$ | 0 | 0 | 0 |
| Vestibular input | $v_{\text{vst}}$ | – | 0 | – |
| Obs. noise for vestibular input | $\sigma_{\text{vst}}$ | – | 0.05 | – |
| Time const. for pred. err. | $\tau_\epsilon$ | – | – | 0.050 s |

Most parameters are maintained throughout all computer experiments. Deviations from these parameters are listed in the respective experiment description. The value $\nu_m = -1$ in location-indexed experiments relates to self-motion. For $D = 2$ spatial dimensions, $v_{\text{self}} = -2/D = -1$ yields a uniform prior distribution (see Supplementary Note 2, Section 1.3).

second assumption is a modeling assumption: that biological agents might disregard the subtle (and complicated) interactions between the uncertainties of different motion sources and rely on their individual uncertainties, instead. Using the two assumptions we derive a closed-form solution for the posterior variance,

$$\Sigma_{mm} = \frac{\sigma_{\text{obs}}^2}{\tau_s \, \|\boldsymbol{c}_m\|^2} \left( -1 + \sqrt{1 + \frac{\tau_s^2 \, \|\boldsymbol{c}_m\|^2}{\sigma_{\text{obs}}^2} \lambda_m^2} \right) =: f_\Sigma(\lambda_m^2) \, , \quad (5)$$

with $\|\boldsymbol{c}_m\|^2 = \sum_{k=1}^K C_{km}^2$ denoting the vector-norm of the $m$-th column of $\boldsymbol{C}$. This is Eq. (3) of the main text. The plot in Fig. 1i has parameters $\|\boldsymbol{c}_m\|^2 = 4$, $\tau_s = 300$ ms, and $\sigma_{\text{obs}} = 0.05$. By plugging the adiabatic approximation of the variance into the time evolution of $\boldsymbol{\mu}_t$, we arrive at Eq. (2) of the main text (see Supplementary Note 2, Section 3.4 for the derivation).

**M-step.** Using the posterior from the E-step, motion strengths, $\boldsymbol{\lambda}$, are optimized to maximize the likelihood of the observed velocities. This optimization further incorporates prior distributions, $p(\lambda_m^2)$, most conveniently formulated over the squared motion strengths, for which we employ a scaled inverse chi-squared distribution,

$$p(\lambda_m^2) = \mathcal{I}\chi(\lambda_m^2 \, ; \, \nu_m, \kappa_m^2) = \frac{1}{\lambda_m^{(2+\nu_m)}} \exp\left[ -\frac{\nu_m \kappa_m^2}{2\lambda_m^2} - A(\nu_m, \kappa_m^2) \right], \quad (6)$$

owing to its conjugacy to estimating the variance of $s_m$ (this is what $\lambda_m^2$ controls). The prior features two hyper-parameters, $\nu_m$ and $\kappa_m^2$, which give rise to an intuitive interpretation as $\nu_m$ pseudo-observations of average value $\kappa_m^2$. The partition function, $A(\nu_m, \kappa_m^2)$, only serves for normalization. By default, we employ a Jeffreys prior ($\nu_m = \kappa_m^2 = 0$), which is a typical choice as a non-informative prior in Bayesian statistics and promotes a preference for finding simple structures by assigning higher beliefs to small values of $\lambda_m$ (and highest to $\lambda_m = 0$). The only exception is the motion strength assigned to self motion, $\lambda_{\text{self}}$, for which we employ a uniform prior distribution, formally by setting $v_{\text{self}} = -2$ and $\kappa_{\text{self}}^2 = 0$. These choices reflect the a-priori belief that motion components supported by $\boldsymbol{C}$ will usually be absent or small in any given scene—with the exception of self-motion-induced velocity on the retina, which occurs with every saccade and every turn of the agent's head (see Supplementary Note 2, Section 1.2 for the formal calculation of the M-step).

In the online formulation of EM (see Supplementary Note 2, Sections 2.3 and 3.4 for the derivation of the online EM algorithm and of the online adiabatic inference algorithm which constitutes our model, respectively), these priors give rise to the low-pass filtering dynamics

in Eq. (1) for updating $\lambda_m^2$, with constants

$$\alpha_m = \frac{2}{\tau_s^2 (2 + \nu_m + \tau_\lambda/\tau_s)} \, , \text{and} \quad (7)$$

$$\beta_m = \frac{\nu_m \kappa_m^2}{\tau_\lambda (2 + \nu_m + \tau_\lambda/\tau_s)}. \quad (8)$$

This completes the derivation of the online model for $D = 1$ spatial dimensions. The extension to multiple dimensions is straightforward and provided in Supplementary Note 2, Sections 1.3, 2.3 and 3.4 alongside the respective derivations.

**Preference for simple structures.** The above Jeffreys prior on motion strengths, $p(\lambda_m^2)$, facilitates the discovery of sparse structures. This property is important when a large reservoir of possible motion components in $\boldsymbol{C}$ is considered: the model will recruit only a small number of components from the reservoir. In Supplementary Fig. 2, we demonstrate this ability for the example of the Johansson experiment from Fig. 2b–d by duplicating the shared motion component, i.e., the first two columns in $\boldsymbol{C}$ are all 1's. As Supplementary Fig. 2 shows, the model recruits only one of the two identical components and discards the other. This example of identical components in the reservoir represents the theoretically hardest scenario for maintaining a sparse structure.

**Computer simulations**
Computer simulations and data analysis were performed with custom Python code (Python 3.8, Numpy 1.21, Scipy 1.7, scikit-learn 0.24, Matplotlib 3.4, Pandas 1.3, xarray 0.19). The code has been published on GitHub[88] and supports most of the extensions presented in Supplementary Note 3.

For the numerical simulation, input was drawn with observation noise variance $\sigma_{\text{obs}}^2/\delta t$, at the time points of input frames (every $\delta t$). The drawn input remained stable until the next frame. Between frames, the differential equations for online hierarchical inference were integrated with SciPy's explicit Runge-Kutta method RK45 which adapts the step size. This integration method combines numerical accuracy with a parameterization that is almost invariant to the input frame rate. The default parameters that we used are listed in Table 1. The data shown in the figures is provided in a supplementary source data file.

**Hierarchical motion experiments (Fig. 2)**
For the Johansson experiment, all $K = 3$ dots followed sinusoidal velocities with frequency 0.5 Hz. Horizontal amplitudes were $2\sqrt{\tau_s}$ for all dots; vertical amplitudes were 0 for the outer dots and $\cos(45°) \cdot 2\sqrt{\tau_s}$ for the inner dot. For the Duncker wheel, we set the wheel radius to $R = 1$ and the rotation frequency to 1 Hz. This leads to the hub velocity

$v_{\text{hub},y} = 0$ and $v_{\text{hub},x} = 2\pi \, \text{s}^{-1}$ because the hub must travel $2\pi R$ during one period for slip-free rolling. For the rim velocities, being the derivatives of location, we thus find $v_{\text{rim},x} = v_{\text{hub},x} + R\omega\cos(\omega t)$ and $v_{\text{rim},y} = -R\omega\sin(\omega t)$, with $\omega = 2\pi \, \text{s}^{-1}$. For the simulation, we increased the observation noise to $\sigma_{\text{obs}} = 0.15$ and set $\lambda_m(t=0) = 0.1$ to highlight the gradual discovery of the motion components.

## Structure classification (Fig. 3)

The stimulus data and human responses were released by Yang et al.[17] on GitHub. The experiment is described in detail in ref. 17. There were 12 participants with each participant performing 200 trials. Each trial consisted of three dots moving on a circle for 4 s. Dots had different colors to prevent their confusion, but colors did not convey any information on the dots' roles within the structure. No data was excluded. Trials were generated stochastically from the same generative model that is considered in this work, with uniform probability for each of the four structures (Independent, Global, Clustered, Hierarchical) to underlie the trial. Motion strengths were chosen such that all dots had identical marginal velocity distributions, $p(v_k)$, across all structures—leaving motion relations as the only distinguishing information (see ref. 17, for detailed stimulus parameters and $\lambda$-values of all structures). Like Yang et al.[17], we treated the experiment as one-dimensional ($D = 1$), operating directly on the angular velocities. Noise-free angular velocities were calculated from the circular distance of subsequent stimulus frames, and we set $1/\delta t = 50\,\text{Hz}$ to match the experiment's frame rate.

For presenting the trials to our online inference model, we initialized each of the $\lambda_m$ at its average value (average taken across the ground truth of all structures). At trial end, the model yielded $M = 7$-dimensional $\lambda$-vectors associated with 1 shared component, 3 cluster components (one per possible pair), and 3 individual components (see Supplementary Fig. 3 for example trials). For logistic regression, we calculated 5 features, $T_i$, from $\lambda$, namely:

$$T_1 = \lambda_1 / \sum_m \lambda_m \qquad \text{Does shared motion stand out?}$$
$$T_2 = \max\{\lambda_2, \lambda_3, \lambda_4\} / \sum_{m=2,3,4} \lambda_m \qquad \text{Does one cluster dominate the others?}$$
$$T_3 = \max\{\lambda_5, \lambda_6, \lambda_7\} / \sum_{m=5,6,7} \lambda_m \qquad \text{Does one individual component stand out?}$$
$$T_4 = \lambda_c^2 / \sum_{m=c,\, \text{Ch}_1(c),\, \text{Ch}_2(c)} \lambda_m^2 \quad \text{with } c = \text{argmax}(\lambda_2, \lambda_3, \lambda_4) \qquad \text{Does the strongest cluster dominate its children?}$$
$$T_5 = \lambda_c^2 / \sum_{m=c,\, \neg\text{Ch}(c)} \lambda_m^2 \quad \text{with } c = \text{argmax}(\lambda_2, \lambda_3, \lambda_4) \qquad \text{Does the strongest cluster dominate the 3rd dot?}$$
$$(9)$$

Here, $\text{Ch}_{1,2}(c)$ denote the individual motion components of the two dots within the cluster component $c$, and $\neg\text{Ch}(c)$ denotes the dot not being in cluster $c$. The features were hand-designed based on the reasoning that they may be useful for structure classification. Their most important property is that all information about a trial is conveyed through $\lambda$ as a bottleneck. A multinomial logistic regression classifier was trained with L1-regularization on the feature vectors, $(T_1, \ldots, T_5)$, to classify the ground truth structures of the trials. Then, we fitted the same choice model as ref. 17 to the human choices, but replaced the ideal observer log-probability, $\log p(S | v_{0:T})$, which was used in ref. 17, with the class probability from the trained classifier, $\log p(S | \lambda)$:

$$P(\text{choice} = S) = \pi_{\text{L}} \frac{1}{4} + (1 - \pi_{\text{L}}) \exp\left[\beta\left(\log p(S | \lambda) + b_S\right)\right] / \text{Norm.}, \quad (10)$$

with lapse probability, $\pi_{\text{L}}$, inverse temperature, $\beta$, and biases, $b_S$, for all structures, $S = G, C, H$, relative to the independent structure ($b_I = 0$ by convention). Note that, in contrast to ref. 17, we do not need to consider structure multiplicities here because the features are already symmetric with regard to the three possible cluster assignments. Like ref. 17, we did not apply observation noise to the presented velocities, but maintained a non-zero observation noise parameter,

$\sigma_{\text{obs}}$, for the inference. Observation noise, $\sigma_{\text{obs}}$, and lapse probability, $\pi_{\text{L}}$, were shared parameters for all participants and were fitted jointly via a simple grid search. We obtained $\sigma_{\text{obs}} = 0.04$ and $\pi_{\text{L}} = 4\%$ (compared to 14% in ref. 17). The remaining 4 parameters, $\{\beta, b_G, b_C, b_H\}$, were fitted via maximum likelihood for each participant. All reported confusion matrices and log-likelihoods were obtained by fitting the 4 per-participant parameters using leave-one-out cross-validation. The log-chance level in Fig. 3f is $200 \cdot \log(1/4)$ since each participant performed 200 trials.

## Location-indexed experiments (Figs. 4–6)

To support self-motion, we introduce a column of −1's in $C$ as an additional component, which is connected to all visual velocity inputs and to a vestibular input $v_{\text{vst}}$. In our simulations, the vestibular input is always stationary, but noisy: $v_{\text{vst}} \sim \mathcal{N}(0, \sigma_{\text{vst}}^2)$. The associated self-motion strength, $\lambda_{\text{self}}$, uses a uniform prior (see discussion under Eq. (6)). Perceived velocities are the sum over all-except-self-motion: $v_{\text{perceived}} = \sum_{m \neq \text{self}} C_{*m} \mu_m$.

## Motion-direction-repulsion (MDR) experiments (Fig. 4)

In the MDR experiments with two RDKs, input was modeled as $K = 3$ velocities: two for the two groups of dots, plus the vestibular input. Repulsion angles were estimated from 20 repetitions of 30 s long trials, with $v_{\text{perceived}}$ averaged over the last 10 s of each trial. Error bars from the simulations were too small to be shown in Fig. 4e–g.

In Fig. 4e, the velocities for opening angle, $\gamma$, were given by $(v_x, v_y) = v_0 \cdot (\cos(\gamma/2), \sin(\gamma/2))$ for the first group, with $v_0 = 2\sqrt{\tau_s}$, and $v_0 \cdot (\cos(\gamma/2), -\sin(\gamma/2))$ for the second group. As in Fig. 3 of ref. 36, the repulsion bias was measured with respect to the full opening angle.

In Fig. 4f, increasing contrast of the second group was modeled as dividing the observation noise variance by a factor, $f$, between 0.001 and 10, leading to variance $\sigma_{\text{obs}}^2 / f$ for this group's input. As in ref. 37, the repulsion bias was measured only with respect to the first group's perceived direction. The expressed similarity to experimental data refers to the "2-motion condition" in Fig. 7 of ref. 37.

In Fig. 4g, the velocity of the second group was multiplied by a factor between 0 and 2, and the repulsion bias was measured only with respect to the first group's perceived direction. For a 60° opening angle, we qualitatively replicate the experimental data from Fig. 2a, b in ref. 38. In order to maintain the simulation parameters from previous conditions, we did not attempt to quantitatively match the speed of targets and distractors in ref. 38. A direct quantitative comparison to the human data from Fig. 4b in ref. 36 is difficult because they had measured the point of subjective equality (PSE) to a 90° opening angle for this stimulus condition, finding a 10° bias for the full opening angle.

For the Takemura experiment[39] in Fig. 4h–l, we used $K = 5$ inputs: two inner RDKs, two outer RDKs, and the vestibular signal, which were organized in the motion tree shown in Supplementary Fig. 5a. If not mentioned otherwise, the simulation parameters matched those from the basic MDR experiment in Fig. 4e. The inner stimuli had $v_x = \pm v_0$, and $v_y = v_0$ if non-zero. The outer stimuli had $v_x = 0$, and $v_y = \pm v_0$. The standard deviation of the observation noise of the outer RDKs was divided by factor 6, reflecting that in ref. 39 the outer RDKs covered a three-times larger area and had twice the dot density of the inner RDKs. Each histogram is based on 200 trial repetitions, which use identical initial conditions but different realizations of observation noise, with perceived velocities measured at trial end. The conditions in panels Fig. 4h–l correspond to figure panels 4a, 4b, 6 left, 6 center, 6 right, in ref. 39. Besides transparent motion (i.e., two perceived velocities), Takemura et al. reported also coherent motion (i.e., only one perceived velocity) for the inner RDKs in a fraction of trials. In our computer simulations, we focused only on the biased perception of two velocities.

## Lorenceau illusion (Fig. 5)

For the Lorenceau illusion, we modeled each dot's velocity as a separate input owing to the spatially distributed nature of the stimulus. As in ref. 42, the two groups of 10 dots each oscillated at a frequency of 0.83 Hz. For the oscillation amplitude, we chose $R = 1/2$ (arbitrary units), leading to velocities $v_x(t) = R\,\omega\,\cos(\omega t)$ for the horizontal group and $v_y(t) = -R\,\omega\,\sin(\omega t)$ for the vertical group, with $\omega = 2\pi \cdot 0.83\,\text{s}^{-1}$. As shown in Supplementary Fig. 6, the model decomposes this stimulus into a deeply nested structure comprising self-, shared-, group-, and individual motion. For the noise-free stimulus condition, we used the default simulation parameters. For the condition with motion noise, the observation noise, $\sigma_{\text{obs}}$, of the visual inputs (not the vestibular input) was multiplied by 25.

## Structure-from-motion (SfM) experiments (Fig. 6)

We treat SfM as a location-indexed experiment owing to experimental findings[50,52]. For computer simulations, we model each cylinder as a ring in the x-z-plane, conflating its height into one receptive field (the simulations still run in 2D with x- and y-dimensions being modeled). The outer cylinder had radius $R = 1.5$, and the inner, if present, $R = 1.0$. Normal rotation speed was 90°/s, and fast speed was 135°/s. Velocities were observed at seven equidistant receptive field locations along the x-axis, $x_{\text{RF}} \in \{1.2, 0.8, 0.4, \ldots, -1.2\}$. These correspond to angles, $\varphi_k$, on the cylinders via $x_{\text{RF}} = R\,\cos(\varphi_k)$ with the inner cylinder covering only five RF locations (cf. Fig. 6b). When presenting velocity observations, $v_k$, each RF location $x_{\text{RF}}$ had multiple overlapping $v_k$ (2 for one cylinder, 4 for nested cylinders where they overlapped). The observation noise for velocity inputs, $\sigma_{\text{obs}}$, was multiplied by 20 for the single cylinder-condition and by 30 for the nested cylinders-conditions, reflecting the high local ambiguity when measuring multiple overlapping speeds and directions[89]. For consistency with other simulations, we provided a vestibular input with the same parameters as in previous location-indexed experiments, although this signal plays no computational role in the SfM simulations.

The model's component matrix, $\boldsymbol{C}$, comprised translational self-motion, rotational motions for the outer cylinder, the inner cylinder (only in nested conditions), and shared for both cylinders (only in nested conditions), as well as translational individual components for each $v_k$ (see Fig. 6c, g). Rotational motion is naturally covered by our model as presented in *Radial and rotational motion sources* earlier in Methods and sketched in Fig. 6b. To solve the correspondence problem of overlapping $v_k$, we devised the following assignment process. At every integration time step, $\delta t$, and for every RF location, $x_{\text{RF}}$, keep the previous assignment with probability 0.7, and continue to the next $x_{\text{RF}}$. Else, that is if the assignment is re-evaluated, calculate the Euclidean distance between the model's expected velocities, $\boldsymbol{C}\boldsymbol{\mu}_t$, and the observed velocities, $\boldsymbol{P}_j \boldsymbol{v}_t$, for all permutations, $\boldsymbol{P}_j$, of the overlapping inputs within this RF. Then choose the assignment, $\boldsymbol{P}_j$, that minimizes the Euclidean distance, i.e., the local prediction error, $\boldsymbol{\epsilon}_t$, within the RF. Once all $x_{\text{RF}}$ were processed in this manner, perform the integration of $\partial_t \boldsymbol{\mu}_t$ according to Eq. (2) using the assigned permutations of $\boldsymbol{v}_t$. This completes the model for SfM perception. We note that, since the integration is performed only after all RF assignments have been made, the resulting global assignment process is independent of the order of iterating over the RFs and could, in principle, be performed in parallel and continuous time. The fact that all computations are spatially confined to information within each RF further improves the process's biological plausibility.

For obtaining the switching distribution in Fig. 6e, we performed a 10,000 s long simulation and followed ideas from ref. 90: first we identified a perceptual threshold as the mode of $\{|\mu_t^{\text{rot}}|\,\forall\,t\}$ (the exact value is actually not important). Then we defined two possible percepts which correspond to positive (negative) values of $\mu_t^{\text{rot}}$. A perceptual switch occurred whenever $\mu_t^{\text{rot}}$ crossed the negative (positive) threshold of the other percept. The Gamma distribution was fitted by maximum likelihood.

## Network implementation (Fig. 7)

A detailed derivation of how to implement the online hierarchical inference model in a neural network model is provided in Supplementary Note 4. In the following, we will focus on the specific model parameters used in the simulations of Fig. 7.

For both simulations (the demonstration in Fig. 7e–j and the proposed experiment in Fig. 7k–m), there were $K = 6$ location-indexed input variables in $D = 2$ spatial dimensions. Input was encoded according to the model of area MT presented in Supplementary Note 4, Section 6. Each velocity, $\boldsymbol{v}_k$, was encoded by a population of 192 neurons, with tuning centers organized on a polar grid with $N_\alpha = 16$ preferred directions, and $N_\rho = 12$ preferred speeds (sketched in Fig. 7c for smaller values of $N_\alpha$ and $N_\rho$). Each neuron in each of the $K$ populations thus has coordinates $(n_\alpha, n_\rho)$ describing its preferred direction and speed. To account for the reported bias of MT tuning toward slow speed[67], the density of preferred speeds became sparser for higher speeds, which we modeled in Supplementary Note 4, Eq. (70) by $\mu_\rho(n_\rho) = \rho_{\min} + d_\rho\,n_\rho^{1.25}$, with $d_\rho = (\rho_{\max} - \rho_{\min})/(N_\rho - 1)^{1.25}$, and $\rho_{\min} = 0.1, \rho_{\max} = 8.0$, for neurons $n_\rho = 0, \ldots, N_\rho - 1$. Preferred directions covered the circle equidistantly. The remaining parameters in the tuning function were $\kappa_\alpha = 1/0.35^2$ and $\sigma_\rho^2 = 0.35^2$ for the angular and radial tuning widths, respectively, and $\psi = 0.1\,\text{Hz}$ for the overall firing rate scaling factor. For the network simulations, we increased the frame rate to $\delta t = 1/120\,\text{Hz}$ for the sake of a higher sampling rate on the x-axis in Fig. 7h–j (the simulation software stores firing rates only at the time of frames).

The distributed population comprised 100 neurons. Readout vectors, $\boldsymbol{a}_x$, for all variables represented by this population were drawn i.i.d. from a standard normal distribution, $\mathcal{N}(0, 1)$, for each vector element. Adjoint matrices were calculated numerically to fulfill the required orthonormality conditions (see Supplementary Note 4, Section 4). The low-pass filtering time constant of the prediction error was $\tau_\epsilon = \tau_s/2 = 0.050\,\text{s}$, such that the prediction error could react to changes in $\boldsymbol{\mu}_t$.

The one-to-one population comprised $M$ neurons (or small populations; $M = 8$ for the demo, and $M = 7$ for the proposed experiment), one per function value, $f_\Sigma(\lambda_m^2)$. The proportionality constant for the readout was $f_\Sigma(\lambda_m^2) = 0.001\,r_{1\text{-to-}1,m}$.

Given the parameters and decoding vectors, the simulation software automatically transforms the differential equations of the online inference model into the corresponding neural dynamics, according to the rules stated in Supplementary Note 4, Section 4. Numerical integration of neural dynamics was performed by the same RK45 method used in the previous simulations.

For the demonstration in Fig. 7e–j, inputs were arranged on a ring of radius $R_k = 1$ with angular location $\vartheta_k = 60° \cdot k$ (measured from the x-axis in counter-clockwise direction). Presented velocities were $(v_x, v_y) = (-2\sin(\vartheta_k), 2\cos(\vartheta_k))$ for $t \leq 1\,\text{s}$, $(2\sin(\vartheta_k), -2\cos(\vartheta_k))$ for $1\,\text{s} < t \leq 2\,\text{s}$, and $(2 + 2\sin(\vartheta_k), -2\cos(\vartheta_k))$ for $2\,\text{s} < t$. In the $\boldsymbol{C}$-matrix underlying the network construction, the shared polar visual component was constructed according to Supplementary Note 1, Eq. (6). The shared translational and the 6 individual components were Cartesian.

In the proposed experiment in Fig. 7k–m, all motion components and the input were Cartesian such that input location played no role (formally, we maintained the circular arrangement of the previous network simulation). Input was generated from the model's underlying generative model for a motion tree comprising 1 shared component and 6 individual components. For a given fraction of shared motion, $q$, we set $\lambda_{\text{shared}}^2 = 2^2 q$ and $\lambda_{\text{ind}}^2 = 2^2(1 - q)$. Maintaining constant total squared motion strength, $\lambda_{\text{shared}}^2 + \lambda_{\text{ind}}^2 = 4$, ensures that the (marginal) input velocity distributions are statistically identical across all input locations and all values of $q$. In total, seven fractions,

$q = 1/8, 2/8, \ldots, 7/8$, of shared motion were presented. Per simulation run, each fraction was presented for 10 s, and simulations were repeated for 10 runs. For the subsequent data analysis, the neural responses of only the 2nd half ($5\,\mathrm{s} \leq t$) of the stimulus presentation were considered to avoid potential initial transients. A standard linear regression model (with intercept; `class sklearn.linear_model.LinearRegression`) was trained to decode the correct $q$ from the distributed population's response, $r_{\mathrm{dis}}$, for the fractions $q = 1/8$ and $q = 7/8$. The resulting linear readout (with intercept) was employed to decode $q$ from $r_{\mathrm{dis}}$ for the remaining stimuli in Fig. 7m.

### Reporting summary
Further information on research design is available in the Nature Portfolio Reporting Summary linked to this article.

### Data availability
No new experiment data was produced for this study. The behavioral data from ref. 17 is available with the original publication. The behavioral data for ref. 36 has been digitized by the authors and is included in the software repository: https://github.com/DrugowitschLab/structure-in-motion/blob/main/data/data_Braddick_2002_Fig3C.txt. Source data are provided with this paper.

### Code availability
Computer simulations, data analyses and visualization have been performed with custom Python code which has been released[88] under the BSD 3-clause license and is available online: https://github.com/DrugowitschLab/structure-in-motion.

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

## Acknowledgements

We thank Anna Kutschireiter for valuable discussions and feedback on the theory. This research was supported by grants from the NIH (NINDS U19NS118246, J.D.), the James S. McDonnell Foundation (Scholar Award for Understanding Human Cognition, Grant 220020462, J.D.), the Harvard Brain Science Initiative (Collaborative Seed Grant, J.D. & S.J.G.), and the Center for Brains, Minds, and Machines (CBMM; funded by NSF STC award CCF-1231216, S.J.G.).

## Author contributions

J.B., S.J.G., and J.D. conceived the study; J.B. developed the theory; J.B. performed the computer simulations; J.B. and J.D. analyzed and discussed the data; J.B., S.J.G., and J.D. wrote the manuscript.

## Competing interests

The authors declare no competing interests.
