## [Peer Review File · Nature Communications]

Visual motion perception as online hierarchical inferenceREVIEWER COMMENTS

Reviewer #1 (Remarks to the Author):

I find the work very interesting and an important contribution to the field. The manuscript itself is very thorough with impressive attention to detail including figures and supplementary information. I have only minor and very minor points.

Reference #9 Weiss et al. (2002). A recent publication) demonstrated that a “slow prior” for motion perception may reflect speed-contrast association rather than a distribution of movements (a prior suggested by the original paper). See Rideaux, R., & Welchman, A. E. (2020). But Still It Moves: Static Image Statistics Underlie How We See Motion. *The Journal of Neuroscience*, 40(12), 2538–2552. <https://doi.org/10.1523/JNEUROSCI.2760-19.2020>

Lines 76: “Such decomposition requires knowledge of the scene’s structure, like the presence of a flock and which birds 77 it encompasses”. Pastukhov (2017) demonstrated how knowledge about scene structure influences motion decomposition into rotation and biological motion. See Pastukhov, A. (2017). First, you need a Gestalt: An interaction of bottom-up and top-down streams during the perception of the ambiguously rotating human walker. *Scientific Reports*, 7(1), 1158. <https://doi.org/10.1038/s41598-017-01376-1>

Very minor: line 70 “...motion sources, s, that...” I would put “s” in brackets as initially it looks like a typo until one reads further (but, again, this is as minor as it gets)

Question: when I watch the videos (extra thank you for including them!) certain conditions lead to perceptual switches between alternative (“veridical” vs. “illusory”) interpretations. Can the model exhibit such bistability (or what would be required modifications) and which time-scale would be the main driver of these oscillations?

Reviewer #2 (Remarks to the Author):

This paper presents a hierarchical inference algorithm to extract global and local motion patterns from velocity input, and several examples showing that the algorithm captures human psychophysics data (from previous publications) and motion illusion phenomenology. They further present a neural network model implementation of the algorithm and suggest that the model captures properties of MT neurons.

It does seem that the authors' hierarchical inference algorithm works quite well in decomposing a set of velocity inputs into underlying motion sources in a hierarchical manner. However, it is unclear to me how much a conceptual advance this hierarchical inference algorithm represents. At some level, it seems more of a methodological improvement that could be presented in a more specialized journal.

In addition, the neural network model does not seem biologically plausible to me at all. I'm not aware of biological mechanisms that allow neurons to perform both linear and quadratic integration (Eq. 4). Overall, the model section is very dense and difficult to read.

Regarding the connection to psychophysics, the tasks presented in Fig. 2 are fairly easy tasks, and the illusion described in Fig. 4 (Lorenz illusion, motion reversal) did not really work for me so I'm not sure if it's a strong illusion. I think there are stronger motion illusions (e.g., motion-induced blindness; bistable perception in structure-from-motion; various apparent motion paradigms) that can present stronger tests of their model.

Lastly, the Discussion section also seems to be concerned mainly with specialized technical issues. This reinforces my impression that this paper represents a technical advance in improving hierarchical inference algorithms that can be used for online motion detection, which, while elegant, is not accompanied by sufficient neurobiological relevance to generate a broad appeal.

Response to reviewers

Thank you for your thoughtful reviews. We have addressed all points and think that these changes improved the quality of the manuscript. First we provide an overview of the main changes, before addressing your comments point-by-point below.

- We have included two stronger motion illusions:
 - Center-surround interactions in motion direction repulsion (MDR), replicating experiments from (Takemura et al., *J. Vis.*, 2011). This is presented in Figure 4, together with the previous MDR experiments.
 - Structure-from-motion (SfM), based on a large body of literature, e.g., (Ullman, *Proc. R. Soc. Lond.*, 1979) and (Treue et al., *Vis. Res.*, 1991). This is presented in a new Figure 6 and a new subsection “Object recognition and perceptual switching of nested structure-from-motion displays”.

Notably, these experiments did not require any changes to the model’s core parameters in the computer simulations.

- Adding the two experiments led to moving existing figures. The Lorenceau illusion, which was previously part of Figure 4, has been moved to its own Figure 5. The network model now is Figure 7.
- The new experiments are also included in Methods: subsections for Figure 4 and Figure 6. In the Supporting Information, we added Figures S5 & S7 and extended the subsection 2.5 “Polar coordinates: rotational and radial motion”, to supplement the new experiments. Supplemental Video S4 demonstrates the proposed SfM displays.
- For the speed-dependence of MDR (Figure 4g), we have found experimental work (Benton and Curran, *Curr. Biol.*, 2003) measuring this condition for a 60 degree opening angle. Our model qualitatively replicated the experimental data without any parameter changes. We have added the model’s percept for a 60 degree opening angle to Figure 4g (complementing the previous 30 and 90 degree conditions).
- A paragraph has been added to the Discussion section relating our work to predictive coding.
- We have noticed that referring to our model as an “algorithm” suggested to some readers that the model was merely a means to us, the researchers, for solving a set of equations. We have thus replaced the word “algorithm” by “model” or a related wording, which more accurately reflects that what we describe is indeed a model for how humans and other animals might infer and use hierarchical structure information from motion relations.

All changes are highlighted as **green text** in the revised manuscript, with the exception of trivial changes (typos, changed figure numbers).

Point-by-point response to Reviewer #1

Reviewer comment: *I find the work very interesting and an important contribution to the field. The manuscript itself is very thorough with impressive attention to detail including figures and supplementary information. I have only minor and very minor points.*

Response: Thank you, we appreciate that you enjoyed the read.

Reviewer comment: *Reference #9 Weiss et al. (2002). A recent publication) demonstrated that a “slow prior” for motion perception may reflect speed-contrast association rather than a distribution of movements (a prior suggested by the original paper). See Rideaux, R., & Welchman, A. E. (2020). But Still It Moves: Static Image Statistics Underlie How We See Motion. The Journal of Neuroscience, 40(12), 2538–2552. <https://doi.org/10.1523/JNEUROSCI.2760-19.2020>*

Response: Thank you for pointing us to this interesting research. We are now discussing this work in Methods (line 492). Also, thank you for providing such accurate references!

Reviewer comment: *Lines 76: “Such decomposition requires knowledge of the scene’s structure, like the presence of a flock and which birds 77 it encompasses”. Pastukhov (2017) demonstrated how knowledge about scene structure influences motion decomposition into rotation and biological motion. See Pastukhov, A. (2017). First, you need a Gestalt: An interaction of bottom-up and top-down streams during the perception of the ambiguously rotating human walker. Scientific Reports, 7(1), 1158. <https://doi.org/10.1038/s41598-017-01376-1>*

Response: Thank you, we have added Gestalt perception (including the reference) when discussing the role of spatial arrangement for visual motion perception (line 453).

Reviewer comment: *Very minor: line 70 “...motion sources, s, that...” I would put “s” in brackets as initially it looks like a typo until one reads further (but, again, this is as minor as it gets)*

Response: For consistency of how we refer to the (all other) variables, we’d prefer to keep the uncomely appearance at this specific occasion as is.

Reviewer comment: *Question: when I watch the videos (extra thank you for including them!) certain conditions lead to perceptual switches between alternative (“veridical” vs. “illusory”) interpretations. Can the model exhibit such bistability (or what would be required modifications) and which time-scale would be the main driver of these oscillations?*

Response: In principle, bistability could arise in the model in response to ambiguous stimuli on the level of motion sources, s , and motion structure, λ , when observation noise drives the system into different local optima. In the revised manuscript, we explore another mechanism of perceptual bistability for the example of structure-from-motion displays: solving the correspondence problem with the aim to minimize local prediction errors. See the new Figure 6 and Section “Object recognition and perceptual switching of nested structure-from-motion displays”. Regarding your original question, which likely referred to the Lorenceau illusion, perception of this illusion varies quite a bit across individuals (previously reported noise dependence, your notion of bistability, or not working at all for the other reviewer). In light of that, we prefer to refrain from speculating about the origins of (occasional) perceptual switches for this experiment.

Point-by-point response to Reviewer #2

Reviewer comment: *This paper presents a hierarchical inference algorithm to extract global and local motion patterns from velocity input, and several examples showing that the algorithm captures human psychophysics data (from previous publications) and motion illusion phenomenology. They further present a neural network model implementation of the algorithm and suggest that the model captures properties of MT neurons.*

Response: Thank you, this is an accurate summary.

Reviewer comment: *It does seem that the authors' hierarchical inference algorithm works quite well in decomposing a set of velocity inputs into underlying motion sources in a hierarchical manner. However, it is unclear to me how much a conceptual advance this hierarchical inference algorithm represents. At some level, it seems more of a methodological improvement that could be presented in a more specialized journal.*

Response: We view the presented model as a significant scientific contribution to the cognitive and neurosciences, and only secondarily as a “methodological improvement” (or “technical advance in improving hierarchical inference algorithms”, as the reviewer writes below). In particular, our aim was not to merely introduce an improved algorithm for hierarchical motion structure inference, but instead to provide a model for how humans and animals might perform such inference, and with it explain previous experimental findings relating to human percepts arising from this inference. We are sorry if this wasn't clear from the previous version of the manuscript.

More generally, the perception of structured motion has puzzled scientists for many decades. Yet, motion perception was usually studied in very different experimental setups, each leading to separate theories which typically involved their own set of rules for explaining observed phenomena. Our model offers a unifying umbrella to a wide range of very different experiments (see Figures 2-6), is derived from a normative account of information processing, and offers an online algorithm that integrates a noisy stream of incoming sensory information in real time. The last point is worth emphasizing as it offers a biologically plausible explanation to *how* the brain could solve the complex equations involved in hierarchical Bayesian inference. Thereby, the requirements to a bio-plausible real-time algorithm are

very different from the requirements usually imposed in statistics and computer science (where all data is stored in a big matrix and iteratively processed by an all-knowing central processing unit). The finding in Figure 3 that our online model outperforms such “ideal” models in explaining human perception further corroborates its relevance for cognitive science and neuroscience.

In order for readers of the manuscript to better appreciate our aims, we now refrain from referring to our model as an “algorithm”, but instead call it a “model” (or related) throughout the revised manuscript.

Reviewer comment: *In addition, the neural network model does not seem biologically plausible to me at all. I'm not aware of biological mechanisms that allow neurons to perform both linear and quadratic integration (Eq. 4). Overall, the model section is very dense and difficult to read.*

Response: The assumption that biological neurons can perform both linear and quadratic interactions is widely accepted — at least in computational neuroscience, and neuroscience more generally. On the single-neuron level, linear integration is the standard in biological neural network models, starting with the McCulloch-Pitts neuron from 1943 (a.k.a., the Perceptron). Quadratic integration had been commonly used in modeling before, see e.g. (Salinas and Abbott, PNAS, 1996) and (Beck et al., J. Neurosci., 2011), and has recently been the specific target of experimental work (Groschner et al., Nature, 2022). We have now included these references in the main text (line 344). On the network level, responsiveness to structured motion, such as rotation and expansion, is a well-known property of area MSTd (see lines 60, 378 and 474 for references) which is shared by our model.

In the Results section, we have focused on the scientific implications of the network model. A detailed presentation of the math behind the network and its relation to the online inference model is provided in Supporting Information, Section 5.

Reviewer comment: *Regarding the connection to psychophysics, the tasks presented in Fig. 2 are fairly easy tasks, and the illusion described in Fig. 4 (Lorencean illusion, motion reversal) did not really work for me so I'm not sure if it's a strong illusion. I think there are stronger motion illusions (e.g., motion-induced blindness; bistable perception in structure-from-motion; various apparent motion paradigms) that can present stronger tests of their model.*

Response: Thank you for pointing out the narrow focus of our previously considered examples. As mentioned in the general response, we have now widened our focus by adding models for two additional experiments: center-surround interaction in MDR (Figure 4h-l and lines 240–261) and structure-from-motion (Figure 6 and lines 287–335). For the latter, we furthermore make testable predictions for novel displays with nested cylinders.

We'd like to emphasize that our model was able to replicate many non-trivial perceptual phenomena in these experiments without requiring any changes to the model's core parameters in the computer simulations (see Methods, lines 627–637 and 645–675).

Reviewer comment: *Lastly, the Discussion section also seems to be concerned mainly with specialized technical issues. This reinforces my impression that this paper represents a technical*

advance in improving hierarchical inference algorithms that can be used for online motion detection, which, while elegant, is not accompanied by sufficient neurobiological relevance to generate a broad appeal.

Response: Thank you for noticing that the Discussion section might have been biased towards technical points. This originated from aiming for brevity in the original submission. To highlight the broad neurobiological relevance of our work, we have now added a paragraph discussing the relation to predictive coding (lines 428–440), a theory that currently receives major attention in neuroscience, as evidenced by three recent review papers, which we cite in the revised manuscript: (Walsh et al., Ann. N. Y. Acad. Sci., 2020), (Millidge et al., arXiv preprint, 2021), and (Mikulasch et al., arXiv preprint, 2022). Furthermore, the assignment process, which we use for the newly added SfM experiment, provides a potential bio-plausible solution to the correspondence problem in SfM that relies solely on local computations (discussed in lines 455–457). Together with the proposed experiment in Figure 7k–m, we expect our work to be relevant to a broad neuroscience community — and, of course, to cognitive sciences research (Figures 2-6). We hope that the reviewer agrees with this assessment.

REVIEWERS' COMMENTS

Reviewer #1 (Remarks to the Author):

The authors addressed all my comments.

Reviewer #2 (Remarks to the Author):

The authors have satisfactorily addressed my earlier comments. The inclusion of additional motion perception illusion strengthens the manuscript significantly.

Response to reviewers

Point-by-point response to Reviewer #1

Reviewer comment: *The authors addressed all my comments.*

Response: Thank you very much for serving as a reviewer and for your constructive comments.

Point-by-point response to Reviewer #2

Reviewer comment: *The authors have satisfactorily addressed my earlier comments. The inclusion of additional motion perception illusion strengthens the manuscript significantly.*

Response: Thank you very much for serving as a reviewer and for your constructive comments.